# Emission ratios of trace gases and particles for Siberian forest fires on the basis of mobile ground observations

Anastasia Vasileva[1], Konstantin Moiseenko[1], Andrey Skorokhod[1], Igor Belikov[1], Vladimir Kopeikin[1], and Olga Lavrova[1,2]

[1]A.M. Obukhov Institute of Atmospheric Physics, Russian Academy of Sciences, Moscow, 119017, Russia
[2]Russian Research Institute of Railway Transport, Moscow, 107996, Russia

*Correspondence to:* Anastasia Vasileva (av@ifaran.ru)

**Abstract.** Boreal forest fires are currently recognized as a significant factor in climate change and air quality problems. Although emissions of biomass burning products are widely measured in many regions, there is still lack of information on the composition of wildfire emissions in Siberia, the region known for its severe wildfire activity. Emission ratios (ERs) are important characteristics of wildfire emissions as they may be used to calculate the mass of species emitted into the atmosphere due to combustion of a known mass of biomass fuel. We analyze observations of carbon dioxide ($CO_2$), carbon monoxide (CO), methane ($CH_4$), total nonmethane hydrocarbons (NMHC), nitrogen oxides $NO_X$ (=NO + $NO_2$), particulate matter ($PM_3$), and black carbon (BC) within two forest fire plume transects made by the moving railway observatory during TRanscontinental Observations Into the Chemistry of the Atmosphere (TROICA) expeditions. Slopes in linear regressions of excess levels of the pollutants are used to obtain $ER_{CO/CO_2} = 10 - 15\%$, $ER_{CH_4/CO} = 8 - 10\%$, $ER_{NMHC/CO} = 0.11 - 0.21\%$ ppmC ppmC$^{-1}$, $ER_{NO_X/CO} = 1.5 - 3.0$ ppb ppm$^{-1}$, $ER_{PM_3/CO} = 320 - 385 \frac{\text{ng m}^{-3}}{\text{µg m}^{-3}}$, and $ER_{BC/CO} = 6.1 - 6.3$ µg m$^{-3}$ ppm$^{-1}$ which fall within the range of uncertainty of the previous estimates, being at the higher edge for $ER_{CH_4/CO}$, $ER_{NMHC/CO}$, and $ER_{PM_3/CO}$ and at the lower edge for $ER_{NO_X/CO}$. The relative uncertainties comprise 5–15% of the estimated $ER_{CH_4/CO}$, $ER_{NMHC/CO}$, and $ER_{PM_3/CO}$ and 10–20% of $ER_{NO_X/CO}$, $ER_{CO/CO_2}$, and $ER_{BC/CO}$. The uncertainties are lower than in many other similar studies and associated mainly with natural variability of wildfire emissions.

## 1 Introduction

Boreal forests and the boreal climate zone, located within approximately $50 - 70°$N, have become an object of increasing attention in recent decades. A unique feature of boreal regions is their high sensitivity to global climate changes and ability to provide global climate feedbacks through a variety of mechanisms (Screen and Simmonds, 2010) including the global carbon cycle (Kasischke, 2000; Zimov et al., 2006; Schuur et al., 2008; McGuire et al., 2009). Indeed, current estimates suggest 10–17% of global carbon is stored in vegetation and soil in boreal forests of which two-thirds are located on the territory of Russia (Tchebakova et al., 1994; Shvidenko and Nilsson, 2003). Under predicted climate change scenarios, this terrestrial carbon may be released into the atmosphere as gases (mainly $CO_2$, CO, and $CH_4$) and particulate matter through different ways including forest fires. Rapid release of large stocks of carbon into the atmosphere by biomass burning occurs during

immediate combustion of organic matter as well as by exposing the remaining carbon stocks in the soil to substantial warming and decomposition for years after the fire event (Balshi et al., 2007; Goetz et al., 2007; Myers-Smith et al., 2007; Randerson et al., 2006; McGuire et al., 2009).

In the future, frequency, severity, and spread of boreal fires may increase in response to climate changes resulting in the concomitant increase in atmospheric concentrations of biomass burning products (Kasischke et al., 1999; Kasischke and Turetsky, 2006; Soja et al., 2007; Marlon et al., 2008; Amiro et al., 2009). Such a response is now recognized for the wildfires in north Eurasia (mostly in Siberia) which are considered to be a significant extra-tropical source and the major driving factor of the variability of climatically important atmospheric species in the northern hemisphere (Lavoué et al., 2000; Kasischke et al., 2005; Yurganov et al., 2005; Simpson et al., 2006; Wotawa et al., 2001).

During severe fire seasons, forest fires significantly affect regional air quality, decreasing visibility and causing respiratory problems (see for example Popovicheva et al. (2014) and references therein), and make an appreciable contribution to the regional air pollution (Cheng et al., 1998; Wotawa and Trainer, 2000). In remote regions of Siberia, emissions (local or transported) from boreal forest fires can also be an important seasonal source of reactive species in the lower atmosphere (Vasileva et al., 2011; Chi et al., 2013). As an example, excess in $CO$ and $NO_X$ may alter the atmospheric oxidation capacity via chains of chemical reactions with OH radicals (Seinfeld and Pandis, 1997; Stockwell et al., 2012) and significantly perturb the background chemistry in the atmosphere (Jaffe et al., 1996; Tanimoto et al., 2000, 2002; Val Martín et al., 2006; Singh et al., 2010).

Rapid transport of combustion products in large scale circulation systems occurs when a significant portion of the products are injected into the free troposphere up to several kilometers or, occasionally, into the lower stratosphere (Fromm et al., 2000; Fromm and Servranckx, 2003; Val Martín et al., 2010). In these cases, smoke plumes from north Eurasia are frequently traced thousands of kilometers downwind over the continent by satellite and aircraft observations (Cahoon et al., 1994; Hsu et al., 1996; Spichtinger et al., 2001; Paris et al., 2009) and may be associated with elevated concentrations of $CO_2$, $CO$, $NO_X$, ozone ($O_3$), and aerosol over North America (Bertschi et al., 2004; Bertschi and Jaffe, 2005; Jaffe et al., 2004; Warneke et al., 2009; Singh et al., 2010; Kondo et al., 2011). There is also an evidence of formation of toxic pollutants such as $O_3$ and aerosols in boreal forest fire plumes, although the rate of formation varies broadly with dispersion conditions and original composition of the exhausts (Honrath et al., 2004; Jacob et al., 2010; Bossioli et al., 2012; Jaffe and Wigder, 2012; Alvarado and Prinn, 2009; Arnold et al., 2015).

Atmospheric aerosol particles from biomass fires in Russia can seriously deteriorate the air quality in Europe under suitable meteorological conditions (Saarikoski et al., 2007) and contribute to Arctic Haze events (Stohl, 2006; Stohl et al., 2007; Cubison et al., 2008; Warneke et al., 2009, 2010) changing the radiation budget of the earth surface and the atmosphere in the northern hemisphere high latitudes (Quinn et al., 2007, 2008; Flanner, 2013; Olsen et al., 2015). Particularly, deposition of black carbon on snow and sea ice surfaces decreases their albedo and thereby can have an important effect on the energy exchange in the Arctic (Hansen and Nazarenko, 2004; Kim et al., 2005). According to Generoso et al. (2007), Russian biomass fires in the extreme fire year 2003 contributed about 40–56% of the total BC mass deposited north of 75°N.

The ability of aerosol particles to act as a cloud condensation nuclei causes modifications of the microphysical and optical properties of clouds thus changing the cloud lifetimes and precipitation patterns (see references in Langmann et al. (2009)).

In all of the above mentioned problems accurate estimation of the amount of carbon released from biomass fires into the atmosphere in the form of gases and particles is important and requires knowledge about emission factors (mass of a chemical compound emitted per unit mass of fuel burned) or emission ratios (amount of compounds emitted divided by that of a reference compound) (Wiedinmyer et al., 2006; Soja et al., 2004; van der Werf et al., 2010; Urbanski et al., 2011). In the present study, we provide estimates of the emission ratios (ERs) which may be converted, when necessary, into emission factors (EFs) using either the carbon mass balance method (Ward et al., 1991; Laursen et al., 1992) or linear correlations between ERs and EFs (Friedli et al., 2001). Such a conversion, however, introduces additional uncertainties, so we focus on the estimates of ERs.

Despite the growing scientific attention to wildfires in the boreal zone, data from direct measurements of biomass combustion products in Siberian ecosystems is still very limited (Cofer et al., 1998; McRae et al., 2006; Paris et al., 2009). In the present study, we employ the unique ground-based measurements from TROICA-9 (4–18 October 2005) and TROICA-11 (22 July – 5 August 2007) expeditions along the Trans-Siberian Railway with use of a mobile railway carriage observatory (Elansky et al., 2009). The lab carriage was attached to a passenger train just behind the electric locomotive and equipped with an integrated observation system composed of a large number of precision instruments measuring trace gases, aerosol particles, and meteorological parameters. Each measurement campaign lasted for two weeks, with the train traveling a 10 000 km midlatitude transect of the country, from Moscow to Vladivostok (east route) and from Vladivostok to Moscow (west route). The complex measurements of the chemical composition of the near-surface air in the TROICA expeditions were used in many previous studies (see for example (Kuokka et al., 2007; Vartiainen et al., 2007; Berezina et al., 2014)) as they provide the unique insight into the spatial distribution of various air pollutants and allow to distinguish between different anthropogenic and natural air pollution sources in various geographical regions of north Eurasia.

## 2 Plume crossing episodes

A summary of the two forest fire plumes observed during TROICA-9 and TROICA-11 campaigns (hereinafter referred to as F1 and F2 events or plumes, respectively) is given in Table 1. Both the plumes were observed in Transbaikalia – a mountainous area in the south Siberia east to the Lake Baikal known for its severe wildfire activity. Due to dry weather conditions during winter, fire season in the region usually starts early in spring and can last from April to October (Giglio et al., 2013; Randerson et al., 2012; Sukhinin et al., 2004; Vivchar et al., 2010). The latitudes of observations ($51 - 53°N$) approximately correspond to the southern border of the boreal forest zone. During the plume crossing events, the train route passed through low mountain ridges with relative heights up to 400 m. The region of F1 observation is classified as taiga woodlands with *Larix gmelinii* and *Betula fruticosa* which grow in south-east Siberia, while F2 plume was observed in taiga forest steppe with *Larix gmelinii*, *Larix sibirica*, and *Pinus sylvestris* which grow widely in the midlatitudes of Eurasia (Klochko and Romanovskaya, 2004).

Each of the F1 and F2 plume crossing events is about 200 km long. The train routes in the region of plume observations are shown in Fig. 1–2 along with the locations of active fires detected by MODIS Terra and Aqua satellites (the MOD14A1

and MYD14A1, collection 6, data was downloaded through NASA search engine https://search.earthdata.nasa.gov/) on the day of the plume observation and as long as two days before. The size of the circles indicating fire spots is proportional to fire radiative power (FRP, taken from the MODIS data) while the fill color shows the day of fire detection. Possible origins of the air sampled within the plumes are shown in Fig. 1–2 with HYSPLIT model (Hybrid Single Particle Lagrangian Integrated Trajectory archive data, available at http://ready.arl.noaa.gov/HYSPLIT.php) three-day backward three-dimensional Lagrangian air parcel trajectories started from 50 m a.g.l. at geographical locations of the train along the railway every hour during the time period of each plume crossing event (Stein et al., 2015; Rolph, 2017). For each particular location, ensembles of trajectories were calculated for a range of heights and horizontal shifts relative to the location to assess a combined impact of various uncertainties on backward trajectory calculations (Stohl, 1998). Since all the trajectories in each ensemble follow a similar air transport pattern, we show in Fig. 1–2 only the trajectories arriving at 50 m height and at the exact geographical locations of the train. The trajectories are color coded in gray scale according to the transport time, with the time stamps along the trajectories shown with black circles at 12 h intervals. The train moved from West to East (Fig. 1) and from East to West (Fig. 2) in F1 and F2 events, respectively. In Fig. 1 and 2 one can see active fires burning very close (in 0–12 h of air transport) to the railway within the plume transects (F1 and F2 events) as well as in more distant (more than 24 h) locations (F2 event, eastern part). Most of the fires affecting the measurements were started no longer than a day before the plume crossing event (although the corresponding fire spot circles are mostly overlapped by the circles on the next day).

Calculation of ERs in the F1 and F2 events requires correct assessment of atmospheric background concentrations of the analyzed species which are used commonly as reference levels in the regression analysis of the measurement data and quantify inputs from various distant emission sources, both natural (including wildfires) and anthropogenic. Such the inputs may be important in F2 event because of more intense wildfire activity compared to the F1 event (note the different FRP scales in Fig. 1 and Fig. 2), with many active fires detected within 24–36 h of air transport to the place of F2 observation according to backward trajectories. These fires might contribute to the elevated background levels of CO, NMHC, $NO_X$, $PM_3$, and BC outside the F2 plume seen in Fig. 4 and Table 3 compared to those in the F1 plume. Backward trajectories also suggest that the F1 plume was sampled across the line of dispersion while the F2 plume was sampled along the line starting from the upwind plume margin which is also supported by MODIS true color scenes (not shown, 02:40–02:45 UTC, 04:25–04:30 UTC on 09 October 2005 and 03:00–03:05 UTC, 04:45–04:50 UTC on 01 August 2007 for Terra MOD021KM and Aqua MYD021KM, respectively, downloaded from https://modis-atmos.gsfc.nasa.gov/IMAGES) with visible fire smoke for the day and place of each plume observation. As a result, the levels of the measured biomass burning products west (downwind) to the area directly affected by the F2 plume were somewhat elevated compared to the east (upwind) side of the plume, so we use only the upwind measurements to setup reference state concentrations for the F2 plume. In the F1 event, the whole set of the measurements directly outside the area affected by the fire plume was used to setup the reference state concentration levels.

During the fire plume observations, the air temperature and humidity measured from the lab carriage were $6 - 12°C$ and 40–55% in the F1 event, and $24 - 29°C$ and 30–50% in the F2 event. The weak winds of 0.3–0.5 $m \cdot s^{-1}$ were observed during the train stops within both the plumes, in close agreement with the data from the Mogocha weather station (WMO index 30673). Close inspection of the backward trajectories in Fig. 1 and Fig. 2 shows that the air contaminated by biomass burning

products was continuously transported in a weak regional wind field during at least two days just before the time of each plume observation. Thus, we can safely assume a negligible contribution of the emissions from fires that burned near the railway in the days prior to the F1 and F2 plume observations into the excess (plume minus background) gas and aerosol levels measured within the plumes. The smoke from active fires detected by MODIS near the railway outside the segment of F1 and F2 plume observations was not measured during the TROICA passes, probably because of insufficient dispersion or a difference between the time of burn and the time of the pass, although emissions from these fires might contribute to the elevated background levels around the F2 plume. On the contrary, the low background levels for the F1 plume suggest that there were no significant emission sources affecting the measured air in this event.

Throughout both the plumes, operators smelled and saw white smoke rising from multiple small fires in the forest on the hillsides approximately 1–1.5 km from the railway. This points to the presence of ground smoldering fires that probably were not detected by MODIS (due to their low radiative temperature, disguise by tree crowns or a difference between time of burn and time of satellite overpass) but contributed to high gas and aerosol concentrations in F1 and F2 plumes alongside the active fires detected by MODIS directly near the railroad within the plumes.

The $CO_2$ and $CH_4$ are the long-lived air constituents with high background levels that are presumably much less affected by local and regional emissions, in contrast with CO, $NO_X$, and other biomass burning products. Yet, pronounced diurnal cycles of $CO_2$ and $CH_4$ were observed during TROICA campaigns in warm seasons, with the maximas during nighttime surface temperature inversions associated with accumulation of local emissions (Belikov et al., 2006; Berezina et al., 2014). Nevertheless, no influence of diurnal $CH_4$ variations on the measurements in F1 and F2 plumes was revealed in the present study, probably due to the absence of strong emission sources (wetlands) in the region of observations. The influence of the nighttime accumulation of $CO_2$ is assumed negligible within F1 and F2 plumes which were observed in daytime well after the breakdown of the inversions. Although, some parts of the plumes are suspected for contributions from non-wildfire emissions and therefore have been excluded from the subsequent analyses (see more discussion below).

Given all of the above, we can safely assume that the peak excess levels of all the chemical compounds measured within F1 and F2 plumes have originated from forest fires located directly near the railway and therefore represent composition of a fresh wildfire smoke, with negligible effects of photochemical aging as well as transformation and removal of aerosol particles (Goode et al., 2000; Hobbs et al., 2003; Stohl, 2006; Paris et al., 2009; Alvarado and Prinn, 2009; Hecobian et al., 2011; Kondo et al., 2011; Chi et al., 2013; Saarnio et al., 2010).

## 3 Measurements and instrumentation

The key characteristics of the measurement instruments used in TROICA campaigns are listed in Table 2 where $PM_3$ are particles with aerodynamic diameters less than $3 \cdot 10^{-6}$ m. Nitrogen oxides were measured with TE42C-TL (TROICA-9) and M200AU (TROICA-11) instruments which register chemiluminescent radiation from reaction of NO with $O_3$, with catalytic conversion of $NO_2$ to NO. Methane and NMHC were measured with Horiba APHA-360 instrument using a single flame ionization detector, with separation of NMHC by a selective absorber. The details on the $CO_2$, CO, and $CH_4$ measurements

are given by Belikov et al. (2006). The NMHC mixing ratios were measured in parts per million by carbon (ppmC) while mixing ratios of other gases were measured in parts per million (ppm) or parts per billion (ppb) by volume.

Previous studies show that a significant fraction of volatile organics in a biomass smoke consists of oxygenated compounds (OVOC) (Akagi et al., 2011; Gilman et al., 2015) which in some cases may be measured as NMHC by flame ionization detectors (Trabue et al., 2013). Our measurements of such OVOCs as acetic acid, acetone, ethanol, methanol, methacrolein, and methyl-vinyl-ketone performed with proton transfer reaction mass spectrometer (PTR-MS) during TROICA campaigns (see Timkovsky et al. (2010), for example) show that mixing ratios of all these compounds are within few ppb while NMHC mixing ratios generally reach hundreds of ppb (see Table 3 and Fig. 3–4 below) that is two orders more. Thus we expect small sensitivity of the NMHC analyzer to OVOC in our study.

All gas analyzers were calibrated daily during the route. For calibration, there were used standards provided by D. I. Mendeleyev Institute for Metrology (Russia), Max Planck Institute for Chemistry (Germany), and Earth System Research Laboratory, NOAA (USA). To perform calibration, gas from the standard cylinder was applied to the instrument via a pressure regulator with a proper pneumatic scheme to perform gas supply under atmospheric pressure. The duration of each calibration was approximately 5–10 minutes. The obtained span coefficients were then used for data recalculation and did not exceed the instruments accuracy values provided in their technical specifications. Zero calibrations for the instruments used to measure CO, $CH_4$, and NMHC were performed every 20 minutes using built-in zero scrubbers. For other instruments, zero air generator was used daily for zero calibrations.

To measure $PM_3$, the Dust Indicator and Tunnel System (model 1.411), designed by GRIMM Corporation (Germany), was used. This instrument was calibrated by nephelometer PHAN-A (photoelectric photometer for aerosols) produced in Russia and calibrated by the manufacturer using the methods which are state-approved in Russia (Kopeikin, 2008). Calibrations were performed immediately before and after each train route. To perform the calibration, synchronous measurements by both the instruments were made during approximately 1 month both in urban and rural regions. The proper zero and span coefficients were obtained and then applied to recalculate the measurements made along the train route. Such the calibration include a wide range of aerosol types from various sources which might partly compensate a possible systematic bias in the measurements of biomass smoke aerosol due to specific particle size distribution, chemical composition, and morphology which may influence the $PM_3$ mass density measured by light scattering (Aurell and Gullett, 2013; Yokelson et al., 2007; Nance et al., 1993).

For black carbon measurements, the single-wave (880 nm) aethalometer (model AE-16) was used (Kopeikin, 2007). This instrument was calibrated in Slovenian Institute of Quality and Metrology (www.siq.si) before the train route. The obtained span coefficient $1.06 \pm 0.16$ was applied to recalculate the data. The accuracy of the single-wave nephelometer used in years 2005 and 2007 is not as high as that of the modern instruments such as Multi-Angle Absorption Photometer or Multi-Wavelength Absorbance Analyzer (Saturno et al., 2016). Nevertheless, the measurements data provides valuable information on wildfire smoke aerosols in boreal Siberia that are still little studied to date.

All the measurements conducted in the TROICA campaigns were fully automated, with all the data available at the central computer. The stability of the measurement system was controlled by operators who also fixed environment settings and some occasional events (oncoming trains, local anthropogenic activity near the railway, biomass burning and industrial plumes,

weather conditions, e.t.c.) in the electronic diary. This meta database was then used at a preliminary data quality control stage as well as in subsequent data processing when studying particular atmospheric events. Thus, the measurements during extra events (oncoming trains, tunnels, populated areas along the road, train stops) according to the records in the diary are not used in the analysis. No systematic influence of the train speed on the trace gas and aerosol observations is revealed in the present

study as well as in the previous analysis of TROICA measurements (Elansky et al., 2009).

The temporal resolution of the original TROICA data is 10 s. Taking into account a range of the instrument response times (Table 2), we averaged the gas mixing ratios and $PM_3$ concentrations over 60 s intervals for subsequent analysis. The BC concentrations were averaged over 300 s intervals.

## 4   Methods of data analysis

The normalized excess ratio (NER) in a biomass burning plume, $ER_{Y/X}$, of a chemical compound $Y$ related to a reference compound $X$ is estimated as the enhancement, above the background, of $Y$ over that of $X$:

$$ER_{Y/X} = \frac{\Delta Y}{\Delta X} = \frac{Y_{plume} - Y_{background}}{X_{plume} - X_{background}}, \tag{1}$$

where $\Delta X$ and $\Delta Y$ are the excess levels (mixing ratios for gases and mass concentrations for aerosols) of the compounds. In fresh plumes, which do not undergo significant chemical and physical transformations of the initial emissions, the NER is an

emission ratio (hereinafter referred to as ER) which may be used to derive emission factors to estimate the mass of the products emitted into the atmosphere when combined with the estimated mass of a fuel consumed. We assume that the NERs estimated with formula (1) in this study may be safely used as emission ratios because we expect that peak $\Delta X$ and $\Delta Y$ come from fires that burned directly near the measurements route.

The $ER_{Y/X}$ in formula (1) is estimated from the slope of linear regression of $\Delta Y$ on $\Delta X$ (Yokelson et al., 1999). According

to a number of studies (Yokelson et al., 1999; Le Canut et al., 1996; Andreae and Merlet, 2001; Guyon et al., 2005; Keene et al., 2006), forcing to zero an intercept term of the linear regression, as stated by (1), can significantly reduce the uncertainty of the resulting ER estimate when the background levels of $X$ and $Y$ are accurately estimated. For F1 and F2 plumes, average background mixing ratios are estimated with the measurements conducted just before and after the plumes (see discussion above) with additional constraints on the upper limits of the measured CO and $NO_X$ to exclude any small scale perturbations

caused by local anthropogenic emissions along the railway (Table 3).

We use CO as the reference compound $X$ in (1) as it shows good correlation ($R^2 > 0.70$) with all the measured species within F1 and F2 plumes. Such the choice in our study is preferable compared to $CO_2$, the other frequently used reference compound, as correlations of the measured species with $CO_2$ appeared to be substantially smaller. High correlations between $\Delta Y$ and $\Delta CO$ in F1 and F2 events could point to the high input of biomass burning products from smoldering combustion

process characterized by relatively high emissions of CO, $CH_4$, NMHC, and particulate matter (Ward et al., 1992; Laursen et al., 1992; Andreae and Merlet, 2001; Hobbs et al., 2003). Noting that many studies provide ERs on the basis of $CO_2$ which accounts for more than 90% of carbon released into the atmosphere from biomass burning, our estimates of $ER_{CO/CO2}$

provide a basis for recalculating CO-based ERs (see, for example, (Le Canut et al., 1996)) to compare the results presented here with other published data, as well as to estimate emission factors for their implementation in current emission models (Yokelson et al., 1999). We also provide $ER_{PM_3/CO}$ and $ER_{BC/CO}$ as the ratios of mass concentrations and the ratios of aerosol mass concentrations to CO volume mixing ratios, respectively, for easy comparison with other studies. The CO volume mixing ratios were converted into mass concentrations with the use of the ideal gas law by utilizing simultaneous measurements of air temperature and pressure along the TROICA route.

Additionally, a modified combustion efficiency (MCE) was estimated on the basis of average $ER_{CO/CO_2}$ for each plume:

$$MCE = \frac{1}{1 + ER_{CO/CO_2}} \tag{2}$$

Formula (2) is widely used to approximate combustion efficiency – the molar fraction of carbon emitted in the form of $CO_2$ in the total amount of carbon emitted from biomass burning including both gaseous phase and particulate matter (Le Canut et al., 1996; Yokelson et al., 1999; Goode et al., 2000; Hobbs et al., 2003).

The MCE is a useful index used to assess relative contributions from flaming and smoldering combustion processes to the measured abundances of species, as well as to compare results of different studies considering large differences between EFs for different types of combustion. Usually emissions of CO, $CH_4$, most of NMHC, and $PM_3$ are higher during smoldering combustion, while emissions of $CO_2$, $NO_X$, and BC are higher during flaming which is therefore associated with higher MCE (Laursen et al., 1992; Ward et al., 1992; Nance et al., 1993; Le Canut et al., 1996; Yokelson et al., 1996, 1999; Goode et al., 2000). Since $CO_2$ and CO together contain over 95% of carbon emitted from biomass burning, the difference between real combustion efficiency and its approximation (MCE) is typically only a few percent.

The analysis of Cantrell (2008) showed that using linear least squares approach to calculate the model slope may produce irrelevant results when both variables are measured with significant noise. In this case, some kind of error-in-variable model would be preferable to account for measurement error in independent variable ($\Delta X$ in our case).

In the present study, we calculate emission ratios for each measured compound with three different linear regression approaches. Two of them use essentially the same standard linear least squares regression algorithm based on singular value decomposition implemented in Linear Algebra PACKage, LAPACK, (Anderson et al., 1999), with $ER_{Y/X}$ estimated as: a slope in linear regression with $Y$ as a dependent variable ($ER_1$) and an inverse of the slope in linear regression with $X$ as a dependent variable ($ER_2$). For algorithms that properly account for uncertainties in both variables, $ER_1 = ER_2$. It is shown below that the latter is not the case in present study, as both $X$ and $Y$ model variables are subject to appreciable (and unknown) amount of uncertainty due to intrinsic inhomogeneity of the emission source as well as varying rates of irreversible mixing with the surrounding air during the atmospheric transport. This problem is addressed in present study by estimating $ER_{Y/X}$ with a third approach ($ER_3$) based on a weighted orthogonal distance regression based on a modified trust region Levenberg–Marquardt algorithm implemented in ORthogonal Distance PACKage, ORDPACK, which accounts for uncertainties in both $Y$ and $X$ variables (Boggs and Rogers, 1990). The weights for each measurement data $(X_i, Y_i)$ pair are then calculated as inverse standard variances of $X_i$ and $Y_i$. The variances include standard deviations of 10 s data values around 60 s averages (the main part) and the measurement uncertainties from Table 2 (a substantially minor part) summed in quadrature.

Trial calculations do not allow to select a particular regression method (of the three methods described above) as the best candidate for ER estimates on the basis of visual inspection of the residual charts. Hence, we calculate the resulting estimates of ER ($ER_{avg}$) for each compound as averages of the slopes from three regression approaches, with standard uncertainties ($\delta ER_{avg}$) calculated according to Bell (1999):

$$ER_{avg} = \frac{1}{3}(ER_1 + ER_2 + ER_3), \tag{3}$$

$$\delta ER_{avg} = \sqrt{U_i^2 + U_{ii}^2}, \tag{4}$$

$$U_i = \frac{1}{n}\sqrt{\sum_{k=1}^{n} \delta ER_k^2}, \ U_{ii} = \frac{1}{\sqrt{n(n-1)}}\sqrt{\sum_{k=1}^{n}(ER_k - ER_{avg})^2}, \tag{5}$$

where $n = 3$ and $(ER_k, \delta ER_k)$ are the model slopes and their uncertainties estimated with three different approaches used in the present study. The implemented method provides a conservative estimate of $\delta ER_{avg}$ as far as covariances among the three algorithms are neglected. Henceforth, for convenience we refer to the corresponding averages given by (3) and (4) as $ER$ and $\delta ER$, correspondingly.

For comparison with other studies, conversion of units is performed, when necessary, with the data provided in original publications. Specifically, $CO_2$-based ERs (Cofer et al., 1989, 1998) are converted to CO-based through dividing by $ER_{CO/CO2}$, with the relative uncertainties summed in quadrature. The EFs (Laursen et al., 1992; Goode et al., 2000; Andreae and Merlet, 2001; Akagi et al., 2011; Urbanski et al., 2009) are converted to ERs following Andreae and Merlet (2001):

$$ER_{Y/X} = \frac{EF_Y\, MM_X}{EF_X\, MM_Y}, \tag{6}$$

where $EF$ (g kg$^{-1}$) is the emission factor, and $MM$ (g) is the molecular weight. The $MM_{NOx}$ is set equal to 30 and 42.8 g for publications in which $NO_X$ was assumed to consist of NO by 100% (Goode et al., 2000; Andreae and Merlet, 2001; Akagi et al., 2011) and by 70–90% (Laursen et al., 1992; Pirjola et al., 2015), respectively.

The $ER_{NMHC/CO}$ (ppmC ppmC$^{-1}$) is calculated from the $ER_{NMHCi/CO}$ (ppmv ppmv$^{-1}$) for individual NMHC compounds using the relation:

$$ER_{NMHC/CO} = \sum_{i=1}^{N} N_{Ci} ER_{NMHCi/CO}, \tag{7}$$

where $N_{Ci}$ is the number of carbon atoms in the $i^{th}$ NMHC compound (NMHC$_i$), $N$ is the number of NMHC compounds measured in the cited study, and $ER_{NMHCi/CO}$ are either provided in the cited study (Friedli et al., 2001) or calculated from the $EF_{NMHCi}$ and $EF_{CO}$ provided in the cited study (Laursen et al., 1992; Urbanski et al., 2009; Akagi et al., 2011) using the relation (6) with CO as $X$ and NMHC$_i$ as $Y$. The choice of the unit of measure for $ER_{NMHC/CO}$ in the present study is related

to the technique used to measure NMHC as well as with the fact that molecular weight of a NMHC compound is related to its chemical efficiency via thermal and photochemical processes leading to the formation of oxidation products and $O_3$ (Friedli et al., 2001). Thus, more heavy and "chemically efficient" NMHC compounds contribute more to the $ER_{NMHC/CO}$ values reported below.

When MCE is not provided in an original publication, it is calculated using $ER_{CO/CO2}$ and formula (2) from the present study.

  The EFs for particulate matter are converted into ERs ($\frac{\text{ng m}^{-3}}{\text{μg m}^{-3}}$) via dividing $EF_{PM3}$ by $EF_{CO}$. This approach is justified by the relation from Laursen et al. (1992) for EF estimates on the basis of carbon mass balance approach: $EF_X = F_C\,1000\,\frac{C_X}{C_T}$, where $EF_X$ (g kg$^{-1}$) is the emission factor for a compound $X$, $F_c$ is the mass fraction of carbon in the fuel, 1000 is the mass

conversion factor (kg to g), $C_X$ (ng m$^{-3}$) is the excess mass concentration of $X$ in biomass burning plume, and $C_T$ (μg m$^{-3}$) is the excess mass concentration of carbon in the plume in form of gases and particulate matter. Assuming $F_C$ and $C_T$ to be constant in a plume (or a series of plumes), we obtain the relation $\frac{EF_Y}{EF_X} = \frac{C_Y}{C_X} = ER_{Y/X}$.

## 5 Results and discussion

Time series of gas mixing ratios and particle mass concentrations measured within F1 and F2 forest fire plumes are shown in

Fig. 3–4 along with the estimated background levels of the measured species plotted for the period of plume crossing. The $NO_2/NO_X$ ratio shown in Fig. 3c and Fig. 4c as an indicator of a "photochemical state" of the plumes reveals that about 80–95% of $NO_X$ in the plumes is in the form of $NO_2$. The high relative fraction of $NO_2$ in $NO_X$ is also reported for other fresh boreal forest fire plumes, which is probably due to rapid NO to $NO_2$ conversion by photochemical oxidation (Laursen et al., 1992; Nance et al., 1993). The highest concentrations in F1 and F2 events were measured during the train stops at

railway stations (Fig. 3–4). Such local episodes of strong anthropogenic contamination are expected to introduce outliers in the regression analyses whose final effect on the inference may be significant. The perturbing effect of local anthropogenic emissions was suppressed through additional filtering of the original data based on some characteristic chemical signatures of the air subjected to local anthropogenic contamination. Namely, the data samples with $\Delta CO > 1.3$ ppm for both the plumes, as well as with $\Delta NO_X > 2.5$ ppb and $NO_2/NO_X < 0.82$ for F1 and $\Delta NO_X > 3$ ppb and $NO_2/NO_X < 0.75$ for F2, were

excluded from the analysis.

  Beyond the data segments corresponding to the train stops described above, peak excess levels within both the plumes were observed near the locations of active fires detected on the same day close to the railway (119.5°E–120.5°E for F1 and 111°E–112°E for F2, compare longitudes in Fig. 1–2 and Fig. 3–4). This supports the above-stated assumption about the dominant contribution of fresh fire smoke to the measurements.

In the remaining parts of the plumes (118.5°E–119.5°E for F1 and 109.5°E–111°E for F1), the measured excess levels of biomass burning products are still much higher than the background levels, although their origins need special discussion. Throughout both the plumes, operators saw and smelled white smoke rising from many small ground fires in the woods on the hillsides, with the smoke filling the observable area. This indicates the presence of fire emission sources directly near the

railroad throughout the whole plume transect in each event, although the fires were probably too small or obscured by tree crowns and therefore were not detected by MODIS. According to Fig. 1, there were no distant emission sources within three days of air transport according to HYSPLIT backward trajectories that could contribute to the measurements in the F1 plume (which is also supported by low background levels of the measured species for the F1 event). Thus we conclude that contribution

of emissions from local fires that burned near the railway on the day of observation was dominant for the measurements in the F1 plume. In Fig. 2 we see many fires between $112°E$ and $114°E$ detected by MODIS during the day of the F2 plume observation and in the previous day as well. Some of these fires were located directly near the railway but were not measured by TROICA (probably due to a time mismatch or insufficient dispersion). The more distant fires located between $112°E$ and $114°E$ within 24–36 h of air transport to the measurements route according to HYSPLIT backward trajectories could contribute

to the elevated background levels for the F2 event. These fires also could contribute to the excess levels measured in the F2 plume segment between $109.5°E$ and $111°E$. In fact, this segment is the only part of the analyzed plume crossing transects F1 and F2 which can be suspected of some appreciable contamination by aged fire smoke, although the latter is not supported by further analysis of $ER_{NOx/CO}$ variations.

In Fig. 3–4 one can see the substantial and simultaneous increases in CO, NMHC, and $NO_X$ mixing ratios and aerosol

concentrations within F1 and F2 plumes compared to those in the ambient air. For long-lived gases $CO_2$ and $CH_4$ with high background levels in the atmosphere the relative excess levels are much smaller reaching as much as 5–10% of the background levels. Variations of the excess levels of all the measured gases and particulate matter are generally well correlated with each other within the plumes, thus supporting the notion of their common emission source. The few exceptions are discussed further.

Thus, Fig. 3 shows a distinct decrease in all excess mixing ratios and mass concentrations during 03:15–04:30 UTC which

corresponds to the railway ascend from 550 to 800 m a.s.l. when the railway crossed a mountain ridge. At the top of the ridge (04:00–04:35 UTC), correlation between the measured concentrations of different compounds is very low, therefore we completely exclude the corresponding data segment from further analysis. Before the top of the ridge (02:50–04:00 UTC), correlation of $CO_2$ with every other measured compound (for example, see gray crosses in Fig. 5d and Fig. 5f, respectively) is also very low while correlation of $CH_4$, NMHC, $NO_X$, and $PM_3$ with CO ($R^2 > 0.7$, Table 5) is as high as in the remaining

part (04:35–06:30 UTC) of the F1 plume. This feature suggests that $CO_2$ observations were influenced by emissions from a non-fire source during that time, therefore we do not report $ER_{CO/CO2}$ for 02:50–04:00 UTC in the F1 plume. We also do not report $ER_{BC/CO}$ for that period because BC shows very low correlation with CO and $CO_2$.

In Fig. 4a two broad $CO_2$ peaks in the western F2 plume part are observed during 04:00–04:20 UTC and 05:00–05:10 UTC, accompanied by short-term fluctuations of $CH_4$ and NMHC (Fig. 4b), as well as an increase in $NO_X$ during 04:00–04:20 UTC

(Fig. 4c). The absence of coinciding increases in CO and $PM_3$ for those periods suggests a non-fire source of these fluctuations, and the diary records indicate the train passage through a town during 04:00–04:20 UTC and a rural settlement during 05:00–05:10 UTC. Since these $CO_2$ peaks strongly affect the CO–$CO_2$ ratios for a large F2 plume part from 03:40 UTC till 05:20 UTC, we do not report $ER_{CO/CO2}$ value for that period. We also do not report the $ER_{BC/CO}$ for the same period because of the low correlation between BC and CO, also probably due to the anthropogenic contamination. The correspond-

ing short-term variations in $CH_4$, NMHC, and $NO_X$ produce outliers in scatter plots in Fig.6a–c (black crosses) which were excluded from the regression analysis.

Given all of the above, one can see that the continuously changing environment of the measurements from the moving platform results in appreciable variations in excess mixing ratios and correlation between the major biomass burning products,

CO and $CO_2$ (as well as between $CO_2$ and other measured compounds that are correlated with CO in this study). These variations, associated with changing surface heights in a mountainous region, as well as with non-fire emission sources, as shown above, interfere with the fluctuations in the measured concentrations attributed to local forest fire emissions. To deal with the heterogeneity in the measurements conditions, we split each of the F1 and F2 plume crossing episodes into two time intervals (parts, or segments, see Table 4) for further analysis according to the observed differences in excess mixing ratios and

the rate of correlation between CO and $CO_2$. The correlation is high during F1-2 and F2-1 ($R^2 > 0.9$), decreased during F2-2 ($R^2 = 0.7$), and low during F1-1 ($R^2 = -0.24$) plume parts, as shown in Table 5.

Scatter plots of excess gas and aerosol levels versus excess mixing ratios of CO and $CO_2$ in F1 and F2 plumes are shown in Fig. 5–6, along with the regression lines for each regression method and plume segment. The final estimates of ERs values ($ER_{avg}$) and corresponding standard deviations ($\delta ER_{avg}$) calculated with formulas (3)–(5) for each plume part from Table 4

are shown in Table 5. Here three sources of uncertainty in the derived estimates are considered: internal variability of the measurement data with the uncertainties $\delta ER_1$, $\delta ER_2$, and $\delta ER_3$ estimated with the particular regression procedure, variability of the ER estimates due to specific choice of the regression model (estimated as $U_{ii}$ with formula (5)), and variations of the ERs between different plume parts within each plume. Herewith the term "uncertainty" means the precision of a model estimate as well as natural variability of the estimated quantity because both these meanings are closely related in the present study.

All the uncertainties in Table 5 represent the range of possible variability of the final ER estimates at 68% level of confidence assuming a normal distribution of the ERs around the estimated values (a common assumption for all the studies reporting ER or EF estimates). The corresponding correlation coefficients ($R^2$) for various $X$ and $Y$ variables are shown in Table 5 below the ER block. The $R^2$ values are high for both trace gas ($R^2 > 0.7$) and gas–particle ($R^2 > 0.5$) correlations.

From Table 5 one can see that the estimated average $ER_{CO/CO2}$ is $15.2 \pm 0.7\%$ for F1-2 and $10.0 \pm 0.6\%$ for F2-1 plume

parts, with the relative uncertainties about 5% of the $ER_{CO/CO2}$ coming mainly from the internal variability of the measurements. As it follows from the laboratory study of Yokelson et al. (1996), the estimated MCE $= 0.91 \pm 0.05$ suggests that a mixture of emissions from flaming and smoldering combustion was sampled within the F2-1 plume part, while the MCE $= 0.87 \pm 0.04$ for F1-2 indicates the dominance of smoldering. For real wildfires, the relationship between visually observed combustion type and MCE may not be so explicit (for example, see Ward et al. (1992), Cofer et al. (1998), and Pirjola

et al. (2015)). Yet, admitting the insufficient amount of a priori information, we retain hereafter the terms "smoldering" and "mixed" in our generic classification of the biomass plumes based solely on MCE values, as a particular combustion regime has an important impact on the chemical composition of the plumes. Indeed, the reported lower $ER_{CO/CO2}$ for the F2-1 plume part may be due to more severe burning conditions observed in summer compared to those observed in autumn within F1-2 plume part, with more flaming combustion during F2-1 producing more $CO_2$ and less CO.

The estimated average $ER_{CH4/CO}$ are quite stable within the plumes, being slightly higher (at 68% confidence level) for the F2 plume (9.7–9.9%) compared to the F1 plume (8.1–8.4%), with the relative uncertainties about 5% of the ERs. The relatively high uncertainty in $ER_{CH4/CO}$ for the F2-2 plume part (15% of the average) is most probably due to the accumulation of $(X_i, Y_i)$ data points in the lower left part of the plot in Fig. 6a which affects the results of individual regression methods but not the average $ER_{CH4/CO}$.

The estimated average $ER_{NMHC/CO}$ is also higher for the F2 plume (0.16–0.21 ppmC ppmC$^{-1}$) compared to that for the F1 plume (0.11–0.12 ppmC ppmC$^{-1}$), with the relative uncertainties of 4–9% coming mainly from internal variability in the measurements. The observed increase in $ER_{NMHC/CO}$ may be partially explained by a decrease in $\Delta$CO, as some previous studies show that excess mixing ratios of most NMHC in fire plumes decrease when MCE increases (Laursen et al., 1992; Yokelson et al., 2011; Burling et al., 2011). Yet, the measurements by Yokelson et al. (1997) and Goode et al. (1999) showed that, for example, light unsaturated hydrocarbons $C_2H_2$ and $C_2H_4$ did not correlate with either $CO_2$ or CO. Another possible reason is that smoke from flaming combustion in the F2 plume contained more heavy hydrocarbons with higher number of carbon atoms in their molecules which contributed more to the total NMHC mixing ratios measured in TROICA expeditions on the carbon basis. The effect of smoke aging in the F2 plume is a less likely reason of the increasing $ER_{NMHC/CO}$ because of the high correlation ($R^2 > 0.9$) between the measured $\Delta$NMHC and $\Delta$CO (as CO has a lifetime of several months in the atmosphere), as well as the low variability in the estimated $ER_{NOx/CO}$ in the F2 plume discussed below. Noting the above-given considerations, we associate the extremely high $ER_{NMHC/CO} = 0.21 \pm 0.01$ ppmC ppmC$^{-1}$ estimate with fresh biomass burning emissions.

The estimated $ER_{NOx/CO}$ are about 1.7 ppb ppm$^{-1}$ for the F1 and 3.0 ppb ppm$^{-1}$ for the F2 plume, with the relative uncertainties up to 10–20% coming mainly from scattering of the measurement data whereas the differences among the ERs obtained with each particular regression method are relatively small. The increase in $ER_{NOx/CO}$ with increasing MCE between F1 and F2 plumes agrees with the laboratory study of Yokelson et al. (1996), although in wildfire plumes NO$_X$ do not always increase with MCE (Laursen et al., 1992). Higher uncertainty of $ER_{NOx/CO}$ compared to ERs for other gases in our study can be explained by substantial variability in wildfire NO$_X$ emissions which depend on the combustion efficiency, nitrogen content of biomass fuel, and even on the deposition of nitrogen (in form of nitrate and ammonium ions in particulate matter) transported from distant pollutant sources onto the fuel surface (tree leaves), with subsequent volatilization during combustion (Nance et al., 1993). Atmospheric NO$_X$ is also prone to higher variability compared to CO and CH$_4$ because NO and NO$_2$ are involved in chains of photochemical reactions limiting their atmospheric lifetime to several days in the midlatitudes (Seinfeld and Pandis, 1997). Nevertheless, from Table 5 we see that the estimated average $ER_{NOx/CO}$ are very stable within each plume, thus indicating a similar photochemical age of the two plume segments in the events considered, in a close agreement with the results of analyses of Fig. 1–2 and Fig. 3–4 above showing that the peak excess levels of the biomass burning products measured in the F1 and F2 events have been originated most probably from the fires located in the vicinity of the measurement route. Consequently, we can safely assume in our calculations that all the measurements used to derive ERs in our study are heavily dominated by smoke from fresh fire plumes with a negligible average effect of chemical transformations.

The estimated $ER_{PM3/CO}$ varies within 320–385 $\frac{\text{ng m}^{-3}}{\mu\text{g m}^{-3}}$ with the relative uncertainties of 4–8% caused mainly by variability in the measured concentrations which, in turn, may come either from natural variability of fire emissions or from aerosol specific measurement errors. The latter are most probably related to the specific features of biomass smoke aerosol incompletely accounted for during the instrument calibration as pointed above.

The estimated $ER_{BC/CO}$ for the two plumes is about 6.2 $\mu\text{g m}^{-3}\,\text{ppm}^{-1}$ with the relative uncertainties up to 20% coming equally from differences among the regression slopes as well as from the standard uncertainties in the slopes for each particular regression model. An important reason of the observed high uncertainties is a limited number of BC observations (8–15 sample pairs against 30–80 for gases and $PM_3$). Yet, the estimated average $ER_{BC/CO}$ for each plume seem to reflect correctly the linear dependencies between BC and CO shown in Fig. 5f and Fig. 6f.

In the following paragraphs, we summarize the uncertainty and variability in the ER estimates reported in Table 5. In the individual ER estimates the ranges of relative variations $\delta ER_{avg}/ER_{avg}$ comprise 5–15% for $ER_{CH4/CO}$, $ER_{NMHC/CO}$, and $ER_{PM3/CO}$ and 10–20% for $ER_{NOx/CO}$, $ER_{CO/CO2}$, and $ER_{BC/CO}$. The variations come mostly from scattering of data points around the regression lines (via model slope uncertainties $\delta ER_1$, $\delta ER_2$, $\delta ER_3$, see equation (5)) due to natural variability of wildfire emissions since the measurement uncertainties listed in Table 2 are small. In some cases, variations of

the ER estimates from different regression methods ($ER_1$, $ER_2$, $ER_3$) around the average $ER_{avg}$ also contributes to the total uncertainty either due to the limited number of observations (in case of $ER_{BC/CO}$) or due to the scattering of data (in case of $ER_{NOx/CO}$) because different regression methods treat scattering of the observed data points around the model line differently. In total, variations of individual ER estimates reported in this study are generally lower than those reported in other studies (see Fig. 7–8 below) although in the latter case it is often difficult to separate natural variability from the measurement

and analytic uncertainty.

    The variability of the reported $ER_{avg}$ between different plume segments within each plume generally does not exceed the variability $\delta ER_{avg}$ within each plume segment. The exceptions are $ER_{CO/CO2}$ and $ER_{BC/CO}$ for which we do not have enough data, the $ER_{NMHC/CO} = 0.21 \pm 0.01$ ppmC ppmC$^{-1}$ for the F2-2 plume part discussed above, and the $ER_{PM3/CO}$ which varies by 50–55 $\frac{\text{ng m}^{-3}}{\mu\text{g m}^{-3}}$ within each plume. The latter may be due to the incomplete calibration of the $PM_3$ measurement

instrument for biomass smoke aerosol as pointed above, therefore we may suggest using the average $ER_{PM3/CO}$ for each plume (which is about $360 \pm 30$ $\frac{\text{ng m}^{-3}}{\mu\text{g m}^{-3}}$ for F1 and $350 \pm 32$ $\frac{\text{ng m}^{-3}}{\mu\text{g m}^{-3}}$ for F2) to address this issue. In other cases, the absence of statistically significant differences between the ERs estimated for different plume segments within each plume supports the assumption about the common photochemical smoke age throughout the plumes, as well as the acceptably small effect of the changing environment on the observations, as discussed above.

We note finally, that the variability of ERs between F1 and F2 plumes is more pronounced than within each plume, with $ER_{CO/CO2}$ decreasing by about 35%, and $ER_{NOx/CO}$ increasing by about 45% in the F2 plume compared to the F1 plume, probably due to more intensive burning processes related to the F2 plume observed in summer contrary to the F1 plume observed in autumn. The $ER_{CH4/CO}$ and $ER_{NMHC/CO}$ also increase from F1 to F2 event by about 15% and more than 35%, respectively. The increase of $ER_{CH4/CO}$ may be explained by moderate decrease in the observed $\Delta CO$, while the increase of

$ER_{NMHC/CO}$ may be also caused by the changes in chemical composition of NMHC emissions as assumed above. Variations of $ER_{BC/CO}$ and $ER_{PM3/CO}$ between the plumes are not seen probably because of high variations within the plumes.

## 6  Comparison with other published results

### 6.1  Gases

The derived gas ERs are compared against other published estimates for boreal forest fires (Fig. 7). It should be noted that most of the previous studies provide estimates for the region of boreal North America (Northern US and Canada) (Cofer et al., 1989, 1998; Laursen et al., 1992; Simpson et al., 2011; Kondo et al., 2011) and Alaska (Goode et al., 2000; Laursen et al., 1992) on the basis of aircraft observations of predominantly fresh plumes (less than a day after emissions). Contrary, there are only a few relevant studies on boreal Eurasia, which we refer to below.

Paris et al. (2009) reports the $ER_{CO/CO_2}$ values of 7.1% and 4.6% for two forest fire plumes in northeast Siberia in July 2008 sampled from aircraft at heights of 1–3 km a.g.l. in a day after emissions. Pirjola et al. (2015) reported $ER_{CO/CO2}$ from a prescribed burning experiment conducted in southern Finland about 200 km north-west to Helsinki in June 2009. The burning area of about 0.8 ha contained predominantly slash (64%) and humus-layer (32%), with surface vegetation composing only 4%. The highest $CO_2$ concentrations in the smoke near the ground measured with a mobile laboratory during the smoldering phase of the fire exceeded the background level by 80–100 ppm which is several times higher compared to the $\Delta CO_2$ of 10–20 ppm measured in F1 and F2 plumes in the present study (Fig. 5d and Fig. 6d), whereas peak $\Delta CO$ values of 1–3.5 ppm were comparable to peak $\Delta CO$ of 1–1.5 ppm in F1 and F2 plumes. The resulting $ER_{CO/CO2} = 3.2\%$ reported by Pirjola et al. (2015) yields MCE = 0.97 typical for predominantly flaming emissions, although this result was attributed by the authors to smoldering combustion on the basis of visual observations.

Cofer et al. (1998) reported an unusually high $ER_{CO/CO2} = 11.3 \pm 2.7\%$ value (MCE = 0.90) from vigorous crowning (flaming) stages of an experimental fire in Siberia (Bor Island, 60.75°N, 89.42°E; 50 ha of live 20 m high pine forest burned in July 1993, with the fresh smoke measured from helicopter) which is comparable to the $ER_{CO/CO2} = 9.4 \pm 1.0\%$ value for flaming wildfires in Canada and the $ER_{CO/CO2} = 12.3 \pm 1.9\%$ for smoldering boreal logging slash fires in North America, but vastly exceeds the $ER_{CO/CO2} = 6.7 \pm 1.2\%$ for flaming logging slash fires in North America reported in the same study. The $ER_{CO/CO2} = 10.0 \pm 0.6\%$ associated with "mixed" combustion in our study is within the range of uncertainty of the Bor Island flaming experiment value for $ER_{CO/CO2}$. The accompanying $ER_{CH4/CO2}$ and $ER_{NMHC/CO2}$ estimates of Cofer et al. (1998) are consistent with, or even lower than, the typical ERs in flaming related plumes.

The $ER_{CO/CO2} = 8.8\%$ reported by McRae et al. (2006) from helicopter flights over experimental ground fires in Siberian pine forest is in the middle range of the published estimates and is compared to the "mixed" (F2-1 plume part) $ER_{CO/CO2} = 10.0 \pm 0.6\%$ from the present study.

Most of the $ER_{CO/CO2}$ in Fig. 7d fall within the range of 6–16%, with 22 estimates obtained from aircraft measurements of forest fire plumes in Northern US, Canada, and Alaska (Cofer et al., 1989, 1998; Goode et al., 2000; Laursen et al., 1992; Simpson et al., 2011; Urbanski et al., 2009) and two estimates obtained from helicopter observations in Siberia (Cofer et al.,

1998; McRae et al., 2006). The $ER_{CO/CO2}$ from the present study for "mixed" and "smoldering" combustion fall within the middle range (6–16%) of the previous estimates. There are also two outliers not shown in Fig. 7 corresponding to the $ER_{CO/CO2}$ values of 18% and 34% (MCE = 0.85–0.75) related to emissions from smoldering wildfires in Canada and Siberia (Cofer et al., 1998). In the lower right part of the scatter plot in Fig. 7d are the $ER_{CO/CO2}$ values of 3.2% and $4.6 \pm 2.0\%$
from Pirjola et al. (2015) (Finland) and Paris et al. (2009) (Siberia), respectively.

It is important to compare the results of the present study with compilations of EFs for bioclimatic zones made by Andreae and Merlet (2001) and Akagi et al. (2011) as the latter values are often used in wildfire emission models including Global Fire Emissions Database (GFED, www.globalfiredata.org). Although Andreae and Merlet (2001) provide EFs for "extratropical forest" (EXTF) on the basis of the substantial amount of studies, in fact only a couple of them provides reliable data for boreal
Eurasia. The inventory of Akagi et al. (2011) inherits the results of Andreae and Merlet (2001) with the updates available at the time of publication and the EXTF zone separated into "boreal forest" (BORF, high latitudes about $50 - 70°$) and "temperate forest" (TEMF). One can see from Fig. 7d that the $ER_{CO/CO2} = 8.5 - 13.4\%$ from the inventories reside at the top half of the estimates. The $ER_{CO/CO2} = 13.4 \pm 4.9\%$ for BORF is close to the "smoldering" $ER_{CO/CO2} = 15.2 \pm 0.7\%$ from the present study, while the $ER_{CO/CO2} = 8.5 \pm 3.1\%$ for TEMF is close to the "mixed" $ER_{CO/CO2} = 10.0 \pm 0.6\%$.
The $ER_{CH4/CO}$ from different studies in Fig. 7a somewhat decreases with MCE, though not very much, since both CO and $CH_4$ are the products of predominantly smoldering combustion (Nance et al., 1993). Herewith, the $ER_{CH4/CO} = 8 - 10\%$ reported in this study are at the top of the published range, along with the $ER_{CH4/CO}$ for boreal North America attributed to different combustion phases (Laursen et al., 1992; Cofer et al., 1989, 1998; Simpson et al., 2011). The $ER_{CH4/CO}$ of $3.5 \pm 1.2\%$ and $3.9 \pm 0.8\%$ reported by Cofer et al. (1998) for flaming and smoldering stages of an experimental fire in Siberia
are much lower compared to ERs from the present study and are at the bottom of the published estimates, along with the $ER_{CH4/CO}$ of $3.8 \pm 3.6\%$ and $4.3 \pm 2.2\%$ for two fires in Canada (Laursen et al., 1992; Cofer et al., 1998). All the $ER_{CH4/CO}$ from the present study lay within the range of uncertainties of the $ER_{CH4/CO} = 7.7 - 8.2\%$ values from Andreae and Merlet (2001) and Akagi et al. (2011).

The $ER_{NMHC/CO} = 0.12 - 0.21$ ppmC ppmC$^{-1}$ reported in this study are at the top of the range of previous esti-
mates along with the $ER_{NMHC/CO} = 0.18$ ppmC ppmC$^{-1}$ for BORF from Akagi et al. (2011) and the $ER_{NMHC/CO} = 0.21$ ppmC ppmC$^{-1}$ for a forest fire in Alaska from Urbanski et al. (2009). In the middle of the range are the $ER_{NMHC/CO} = 0.08 - 0.09$ ppmC ppmC$^{-1}$ values derived from the sum of EFs for $C_2H_6$, $C_2H_4$, $C_2H_2$, $C_3H_8$, $C_3H_6$, and $C_3H_4$ for two forest fires in Canada and one in Alaska (Urbanski et al., 2009). Not shown (because of the lack of MCE) in Fig. 7b is the $ER_{NMHC/CO} = 0.11$ ppmC ppmC$^{-1}$ estimated with a composite of aircraft observations of $C_2 - C_{10}$ hydrocarbons in four
plumes from vegetation fires in temperate forests of the US (Montana, Colorado) (Friedli et al., 2001). At the bottom of the plot in Fig. 7b are the $ER_{NMHC/CO} = 0.03 - 0.08$ ppmC ppmC$^{-1}$ values derived from the sum of EFs for $C_2H_6$, $C_2H_2$, $C_3H_8$, $C_3H_6$, i-butane $C_4H_{10}$, and n-butane $C_4H_{10}$ for five forest fires in Canada and one in Alaska (Laursen et al., 1992). The observed variations in the $ER_{NMHC/CO}$ estimates are large and associated with natural variability of wildfire emissions of individual NMHC compounds (see Friedli et al. (2001) and references therein), as well as with differences in the measurement
techniques (Rasmussen et al., 1974). Thus, Laursen et al. (1992) and Urbanski et al. (2009) report the measurements of a very

limited number of individual NMHC compounds which can not be directly compared to the comprehensive NMHC measurements employed in the present study but are shown in Fig. 7b because of the deficit of NMHC observations in boreal forest fire plumes. Also, the relation between $ER_{NMHC/CO}$ and MCE is difficult to see in Fig. 7b because of the large differences between the few $ER_{NMHC/CO}$ estimates.

The $ER_{NOx/CO} = 1.6 - 3.1$ ppb ppm$^{-1}$ values reported in this study are at the bottom range of the published estimates, along with the $ER_{NOx/CO}$ of $1.2 \pm 1.7$ and $3.1 \pm 3.2$ ppb ppm$^{-1}$ obtained from aircraft observations of two fires in Ontario (which also have the $ER_{CH4/CO}$ comparable to the results of the present study) derived from Laursen et al. (1992). Other four estimates from Laursen et al. (1992) for fires in Canada and Alaska yield $ER_{NOx/CO} = 11 - 22$ ppb ppm$^{-1}$ with the uncertainties of 100–150% and more. The $ER_{NOx/CO}$ from the present study are also at the bottom of the range of uncertainty

(which is about 100%) of BOR $ER_{NOx/CO} = 6.6 \pm 5.6$ ppb ppm$^{-1}$ derived from Akagi et al. (2011) and the $ER_{NOx/CO} = 7.6 \pm 4.9$ ppb ppm$^{-1}$ values derived from the EFs of Simpson et al. (2011) obtained from airborne measurements of predominantly smoldering fires in Canada in 2008. A distinct outlier in Fig. 7c is the $ER_{NOx/CO} = 33.9 \pm 4.5$ ppb ppm$^{-1}$ of Pirjola et al. (2015) obtained from ground-based observations of predominantly smoldering fire smoke in Finland, which is several times higher compared to the upper limit of other published estimates and was derived by dividing the relatively high

$EF_{NOx} = 2.7 \pm 0.3$ g kg$^{-1}$ by very low $EF_{CO} = 52.1 \pm 2.7$ g kg$^{-1}$. Such the high variability of the published estimates is typical for NO$_X$ emissions and seems to reflect natural variability rather than the uncertainties associated with different methods of measurements and analysis. Thus, within the single study of Laursen et al. (1992), a series of measurements in different fire plumes in Canada yielded $EF_{NOx}$ varying by an order of magnitude. Herewith, the $ER_{NOx/CO}$ from different studies increase with MCE because NO$_X$ and CO are emitted from different (flaming and smoldering, respectively) combustion processes.

Note that the uncertainties in the ERs (where available) shown in Fig. 7 can be as large as 100–200% and more and represent the natural variability of the emissions within a single fire event (Laursen et al., 1992), variability between different fires in a region (Cofer et al., 1998; Simpson et al., 2011), as well as the uncertainties associated with measurement techniques (as in the case of NMHC) and data analysis.

## 6.2   Aerosols

There are only a limited amount of data published on aerosol emissions from boreal biomass fires. We compare our estimates for $ER_{PM3/CO}$ and $ER_{BC/CO}$ with previously published data noting that the results from other studies are actually based on the measurements of particles with aerodynamic diameters less than $2.5 \cdot 10^{-6} - 3.5 \cdot 10^{-6}$ m (PM$_{2.5}$−PM$_{3.5}$). We consider this difference not significant for our quantitative comparison as PM$_{2.5}$ particles contribute most of the total particle mass in fresh biomass burning plumes (Reid et al., 2005; Akagi et al., 2011; Pirjola et al., 2015; Popovicheva et al., 2015).

The $ER_{PM3/CO} = 320 - 385 \, \frac{\mathrm{ng\,m}^{-3}}{\mathrm{\mu g\,m}^{-3}}$ (Fig. 8a) with the standard uncertainty of 4–8% from the present study are at the top of the standard uncertainty ranges (which are 25–85% where available) of the $ER_{PM3/CO} = 196 - 265 \, \frac{\mathrm{ng\,m}^{-3}}{\mathrm{\mu g\,m}^{-3}}$ estimated for three forest fires in Canada (Ontario and British Columbia) and Alaska (Nance et al., 1993; Urbanski et al., 2009). The $ER_{PM2.5/CO} = 122 - 143 \, \frac{\mathrm{ng\,m}^{-3}}{\mathrm{\mu g\,m}^{-3}}$ from Akagi et al. (2011) and Andreae and Merlet (2001), as well as the $ER_{PM2.5/CO} = 35 - 130 \, \frac{\mathrm{ng\,m}^{-3}}{\mathrm{\mu g\,m}^{-3}}$ for other six forest fires in Canada and Alaska (Nance et al., 1993; Urbanski et al., 2009), are 2–10 times lower

compared to the $ER_{PM3/CO}$ reported in the present study. The $ER_{PM2.5/CO} = 557 \pm 92 \frac{\mathrm{ng\,m^{-3}}}{\mathrm{\mu g\,m^{-3}}}$ derived from Pirjola et al. (2015) for a prescribed forest fire in Finland is about 1.5 times higher compared to the values obtained in the present study.

The $ER_{BC/CO} = 6.2 \pm 1.3 \; \mathrm{\mu g\,m^{-3}\,ppm^{-1}}$ from the present study falls within the range of uncertainty of the previous estimates (Fig. 8b). So far the authors know only two studies reporting BC ERs for plumes sampled less than a day after emissions. Thus, Paris et al. (2009) reports the $ER_{BC/CO}$ values of 4.1 and 6.8 $\mathrm{\mu g\,m^{-3}\,ppm^{-1}}$ for two forest fires in Siberia, which are within double standard uncertainties of the ERs from the present study. Chi et al. (2013) estimated the $ER_{BC/CO} = 10 \; \mathrm{\mu g\,m^{-3}\,ppm^{-1}}$ (not shown in Fig. 8b because of the lacking MCE estimate) for forest fires in West Siberia in July 2007, some of which were located very close to the ground measurement site.

The other studies report BC ERs for plumes of several days old. Thus, Warneke et al. (2009) provides the $ER_{BC/CO}$ value of $7 \pm 4 \; \mathrm{\mu g\,m^{-3}\,ppm^{-1}}$ for forest fire plumes originated from Siberia near Lake Baikal and of $10 \pm 5 \; \mathrm{\mu g\,m^{-3}\,ppm^{-1}}$ for agricultural fire plumes originated from Kazakhstan and sampled over Alaska in April 2008, which are within the range of uncertainty of the $ER_{BC/CO} = 8.5 \pm 5.4 \; \mathrm{\mu g\,m^{-3}\,ppm^{-1}}$ obtained by Kondo et al. (2011) for the plumes originated from wildfires in the same geographical areas and sampled at an earlier stage of their evolution similar to the study of Warneke et al. (2009).

The lowest $ER_{BC/CO} = 1.7 \pm 0.8 \; \mathrm{\mu g\,m^{-3}\,ppm^{-1}}$ was obtained by Kondo et al. (2011) for fresh smoldering fire plumes in Canada in summer 2008. The results of Kondo et al. (2011) for flaming ($ER_{BC/CO} = 3.4 \pm 1.6 \; \mathrm{\mu g\,m^{-3}\,ppm^{-1}}$, MCE > 0.95) and mixed ($ER_{BC/CO} = 2.3 \pm 2.2 \; \mathrm{\mu g\,m^{-3}\,ppm^{-1}}$, 0.90 < MCE < 0.95) fire plumes in North America are also lower than, or the bottom edge of, the standard uncertainties of the $ER_{BC/CO}$ from the present study. The highest $ER_{BC/CO} = 21.8 - 29.8 \; \mathrm{\mu g\,m^{-3}\,ppm^{-1}}$ values published for agricultural fires in southern Russia are based on the measurements at the ground site in central Siberia in April 2008 (Chi et al., 2013) and at the Mount Cimone (2165 m a.s.l.) station in Italy in May 2009 (Cristofanelli et al., 2013).

Chi et al. (2013) also provides an overall average $ER_{BC/CO} = 9.3 \; \mathrm{\mu g\,m^{-3}\,ppm^{-1}}$ ($R^2$ = 0.55) for winter air masses measured at the background site in central Siberia since September 2006 till December 2011 that have been previously affected by anthropogenic emissions in the south and southwest Siberia. While Chi et al. (2013) states that their $ER_{BC/CO}$ "is higher than values normally found at rural sites and even at the higher end of the literature range for cities in Asia", the provided value also falls within the range of published $ER_{BC/CO}$ estimates for forest fire plumes.

We should note that most of the $ER_{BC/CO}$ estimates considered above (Kondo et al., 2011; Warneke et al., 2009; Chi et al., 2013; Cristofanelli et al., 2013) are actually the enhancement ratios characterizing the enhancement above a background level of one compound relative to an other in a highly aged plume (of several days old) subjected to substantial dilution and physical or chemical processing. The enhancement ratio is obviously different from the emission ratio characterizing essentially the original chemical composition of an emission plume. Nevertheless, considering the extremely limited number of studies reporting BC and CO emissions from boreal wildfires, we decide to include all the available data into our comparison.

For comparison with Andreae and Merlet (2001), we estimate the $ER_{BC/CO}$ of $5.6 \pm 0.6$ and $6.2 \pm 1.2 \; \frac{\mathrm{ng\,m^{-3}}}{\mathrm{\mu g\,m^{-3}}}$ for the F1-2 and F2-1 plume parts, respectively, which agree with the $ER_{BC/CO} = 5.2 \pm 2.5 \; \frac{\mathrm{ng\,m^{-3}}}{\mathrm{\mu g\,m^{-3}}}$ derived from the above-cited study.

Hence, the published ERs for particulate matter and black carbon from biomass fires in boreal regions vary in a broad range. The most probable reasons for such the strong variability are variations in combustion efficiency of the source fire (which seems to be higher for agricultural fires compared to forest fires) as well as variations in atmospheric dispersion and deposition conditions during the plume transport, as the effects of plume dilution and chemical aging increase with transport time (Kondo et al., 2011). Finally, we conclude that the estimates $ER_{PM3/CO} = 320 - 385 \, \frac{\mathrm{ng\,m^{-3}}}{\mathrm{\mu g\,m^{-3}}}$ and $ER_{BC/CO} = 6.1 - 6.3 \, \mathrm{\mu g\,m^{-3}\,ppm^{-1}}$ reported in the present study fall into the middle of the range of the published estimates.

## 7   Conclusions

In this study, we analyze the time series of ground measurements of the near-surface air chemical composition in two Siberian boreal forest fire plumes to estimate the emission ratios for the primary biomass burning products. In both plumes, a pronounced increase in all the measured species above their background concentrations was observed, with the excess levels of individual compounds well correlated with each other. Each plume transect was about 200 km long and located in the area affected by very weak anthropogenic activity. The amount of the measurement data collected within each plume has proved to be sufficient for reliable statistical inference. Consequently, the derived ER estimates were found to be steady with respect to a particular choice of the regression model and robust to some amount of outliers arising in measurement data due to a range of sampling conditions. The analysis of MODIS active fire detections and HYSPLIT backward trajectories, accompanied by visual observations of many smoldering fires near the train route, has shown that the excess levels of the biomass burning products measured within the plumes in the present study refer to a fresh fire smoke with the negligible average effect of chemical transformations. Consequently, the estimated ERs can be safely assumed to characterize the initial chemical composition of wildfire emissions.

We report the CO-based ERs for $CH_4$, NMHC, $NO_X$, $PM_3$, and BC, as well as CO to $CO_2$ ratios obtained from slopes of linear regression of the excess levels of the species calculated through three different approaches to quantify the effect that different assumptions on errors in the regression variables have on the final estimates. The derived gas ERs are generally stable within the plumes, with the differences between the ERs estimated for different plume segments being statistically insignificant, which supports the general idea of a common fire smoke age throughout each plume, as well as a negligible effect of the changing environment on the measurements.

The estimated gas ERs vary appreciably between the plumes due to the changes in combustion processes manifested via changes in MCE. The high $MCE = 0.91 \pm 0.05$ observed in the F2 summer event probably indicates more intensive burning and flaming combustion compared to the $MCE = 0.87 \pm 0.04$ for the F1 autumn event which may be dominated by smoldering combustion process from fires of lower intensity according to the MODIS data. Consequently, the $ER_{CO/CO2}$ decreases by 35% and $ER_{NOx/CO}$ increases by 45% in the F2 plume compared to the F1 plume, since CO is the product of smoldering combustion while $CO_2$ and $NO_X$ are the typical products of flaming combustion. The $ER_{CH4/CO}$ and $ER_{NMHC/CO}$ also increase from F1 to F2 plume by 18% and more than 35%, respectively, although the $CH_4$, NMHC, and CO are the typical products of smoldering combustion. Such increase in $ER_{CH4/CO}$ value can be explained by a decrease in CO while the

corresponding increase in $ER_{NMHC/CO}$ is probably associated with the accompanying changes in chemical composition of NMHC emissions as well. Compared to the gaseous ERs, the variability of the gas-particle ERs was more affected by the precision of the $PM_3$ and BC measurements, therefore we finally report only one $ER_{PM_3/CO}$ and $ER_{BC/CO}$ value with relatively high total uncertainty for each plume.

The uncertainties in the ER estimates are associated mainly with the variability of wildfire emissions (combustion phase, nitrogen content in the fuel) as well as with the choice of the regression approach as different assumptions on independent variables inevitably affect the final statistical inference. Chemical transformations (photochemical loss of $NO_X$ and oxidation of NMHC) of the initial wildfire emissions during their transport to the measurement route seem to have no effect on the reported average ERs and their uncertainties because of the proximity of fire emission sources to the TROICA route. All

the uncertainties are summed to represent the total variability of each ER estimate which comprises 5–15% of the reported $ER_{CH_4/CO}$, $ER_{NMHC/CO}$, and $ER_{PM_3/CO}$ values and 10–20% of the reported $ER_{NO_X/CO}$, $ER_{CO/CO_2}$, and $ER_{BC/CO}$ values. The resulting uncertainties are generally lower than those reported in many other similar studies. The reported ERs generally fall within the range of variability of the published estimates including those incorporated into some widely used wildfire emission models, although the ER values from the present study are higher compared to most of the previously

published $ER_{CH_4/CO}$, $ER_{NMHC/CO}$, and $ER_{PM_3/CO}$ values and are much lower than most of the previous $ER_{NO_X/CO}$ values.

The authors did not find any definite relation between the visually observed combustion type (smoldering or flaming) and MCE values neither in this study, mainly because of the lack of detailed information on fire state, nor in the previous studies where emissions from experimental fires were attributed to flaming or smoldering combustion on the basis of visual inspections

(Cofer et al., 1998; Pirjola et al., 2015). Thus, we are cautious in using visual observations to attribute fire emissions to a specific combustion type since both flaming and smoldering typically occur simultaneously for naturally burning forest fires. More detailed analysis can not be conducted within the present study as the employed measurement data were not designed originally to study wildfire emissions and the plumes were measured by accident. Nevertheless, the scarcity of information about wildfires in southern Siberia encouraged us to publish the ER estimates with the available measurements which are

unique in that sense.

*Author contributions.*   A. Vasileva designed the study and prepared the manuscript. K. Moiseenko formulated the problem and took an active part in preparation of the manuscript. A. Skorokhod prepared and managed TROICA expeditions. I. Belikov prepared and supported the measurement instrumentation for TROICA expeditions. V. Kopeikin was completely responsible for the $PM_3$ and BC measurements during TROICA expeditions. O. Lavrova conducted the measurements and was responsible for diary observations during TROICA expeditions,

contributing to the description of the analyzed plume crossing episodes.

*Acknowledgements.*   The authors thank Shumsky R. A. for an active participation in designing of the measurement set of the mobile laboratory and controlling its correct work. This study was supported by the Russian Science Foundation (grant no. 14-47-00049), by the Russian

Foundation for Basic Research (grant no. 17-05-00245_A and 15-05-02457-a), and contributes to the Pan-Eurasian Experiment (PEEX) Program research agenda.

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

**Table 1.** Two forest fire plumes observed during TROICA expeditions.

| Plume ID | Date | UTC time (hh:mm) | Local time (hh:mm) | Latitude (deg) | Longitude (deg) |
|---|---|---|---|---|---|
| F1 | 9 October 2005 | 02:50–06:30 | 10:50–14:30 | 53.5 | 118.5–120.5 |
| F2 | 1 August 2007 | 01:25–05:20 | 09:25–13:20 | 51.5 | 109.5–112.0 |

**Table 2.** Instruments used for trace gas and aerosol measurements in TROICA expeditions.

| Specie | Model and manufacturer | Measurement method | Measurement range | Measurement uncertainty | Response time |
|--------|------------------------|--------------------|-------------------|--------------------------|---------------|
| $CO_2$ | LI6262 (LICOR, USA) | non-dispersion infrared spectrometry | 0.2–3000 ppmv | ± 1 ppmv | 10 s |
| CO | TE48S (Thermo Environmental Inc., USA) | non-dispersion infrared spectrometry | 0.05–50 ppmv | ± 0.01 ppmv | 60 s |
| $CH_4$ | APHA-360 (Horiba, Japan) | flame ionization | 0.05–50 ppmv | ± 1% | 60 s |
| NMHC | APHA-360 (Horiba, Japan) | flame ionization with selective adsorption | 0.05–50 ppmC | ± 1% | 60 s |
| NO, $NO_2$ | TE42C-TL (Thermo Electron Corp., USA); M200AU (Teledyne API, USA) | chemiluminescence | 0.05–200 ppbv | ± 1% | 60 s |
| $PM_3$ | Grimm Dust Indicator 1.411 (GRIMM Aerosol Technik GmbH & Co. KG) | 90° scattering light nephelometry | 0.01–15 $mg\,m^{-3}$ | ±5% | 10 s |
| BC | AE-16, (Magee Scientific, Berkeley, USA) | optical attenuation | $0.01 - 10^4\ \mu g\,m^{-3}$ | ± 20% | 300 s |

**Table 3.** Background levels of trace gases and aerosols outside the F1 and F2 plumes.

| Plume ID | $CO_2$ (ppmv) | CO (ppmv) | $CH_4$ (ppmv) | NMHC (ppmC) | $NO_X$ (ppbv) | $PM_3$ ($\mu g\,m^{-3}$) | BC ($\mu g\,m^{-3}$) |
|---|---|---|---|---|---|---|---|
| F1 | 390 | 0.15 | 1.900 | 0.250 | 1.2 | 20 | 1.0 |
| F2 | 365 | 0.24 | 1.755 | 0.255 | 1.7 | 40 | 1.2 |

**Table 4.** Time intervals (plume parts) within F1 and F2 plumes used for the analysis of emission ratios.

| Abbreviation | Date | Time (UTC) |
|---|---|---|
| F1–1 | 09 October 2005 | 02:50–04:00 |
| F1–2 | 09 October 2005 | 04:35–06:30 |
| F2–1 | 01 August 2007 | 01:25–03:40 |
| F2–2 | 01 August 2007 | 03:40–05:20 |

**Table 5.** Average emission ratios (with standard uncertainties) estimated with linear regression and corresponding coefficients of correlation ($R^2$) for excess levels of trace gases and particles in F1 and F2 forest fire plumes.

| | Emission ratios | | | | | |
|---|---|---|---|---|---|---|
| Plume part | $CO$ / $CO_2$ (ppm ppm$^{-1}$ in %) | $CH_4$ / $CO$ (ppm ppm$^{-1}$ in %) | NMHC / CO (ppmC ppmC$^{-1}$ in %) | $NO_X$ / CO (ppb ppm$^{-1}$) | $PM_3$ / CO ($\frac{\text{ng m}^{-3}}{\mu\text{g m}^{-3}}$) | BC / CO (µg m$^{-3}$/ppm) |
| F1–1 | – | $8.1 \pm 0.4$ | $11.5 \pm 1.0$ | $1.8 \pm 0.3$ | $385 \pm 17$ | – |
| F1–2 | $15.2 \pm 0.7$ | $8.4 \pm 0.5$ | $12.4 \pm 0.5$ | $1.6 \pm 0.3$ | $337 \pm 26$ | $6.1 \pm 0.6$ |
| F2–1 | $10.0 \pm 0.6$ | $9.7 \pm 0.2$ | $15.8 \pm 0.6$ | $2.8 \pm 0.2$ | $377 \pm 24$ | $6.3 \pm 1.3$ |
| F2–2 | – | $9.9 \pm 1.5$ | $21.4 \pm 1.0$ | $3.1 \pm 0.4$ | $321 \pm 20$ | – |
| | Coefficients of correlation | | | | | |
| F1–1 | – | 0.95 | 0.94 | 0.74 | 0.94 | – |
| F1–2 | 0.94 | 0.94 | 0.97 | 0.76 | 0.95 | 0.94 |
| F2–1 | 0.92 | 0.98 | 0.96 | 0.87 | 0.94 | 0.80 |
| F2–2 | – | 0.83 | 0.94 | 0.81 | 0.89 | – |

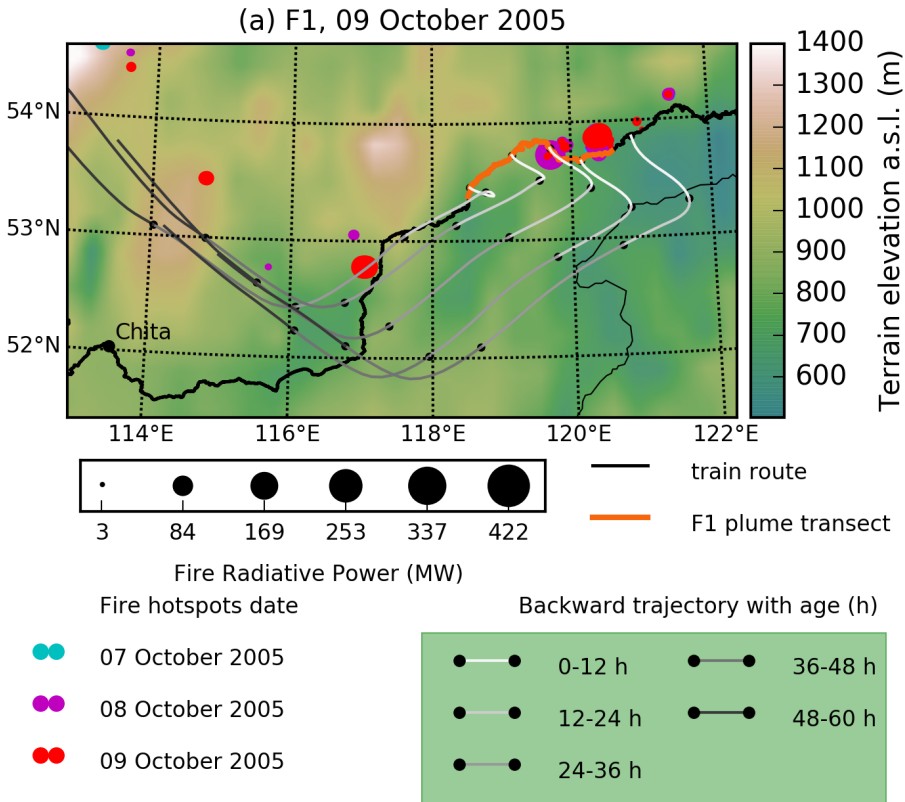

**Figure 1.** Map of the train route with the F1 plume transect location. The train course is from West to East. Circles show active fires detected by the MODIS satellite during the day of plume observation and two days before that and are colored by date and sized by fire radiative power (FRP). Gray lines with open markers show the ensembles of HYSPLIT model backward air parcel trajectories started with hourly time increments along the train route within the plume and the time stamps coded by the number of hours before arrival of an air parcel at the point of observation. Chita is a town with the population of 343 511 (Russian Government Statistical Service, 2016).

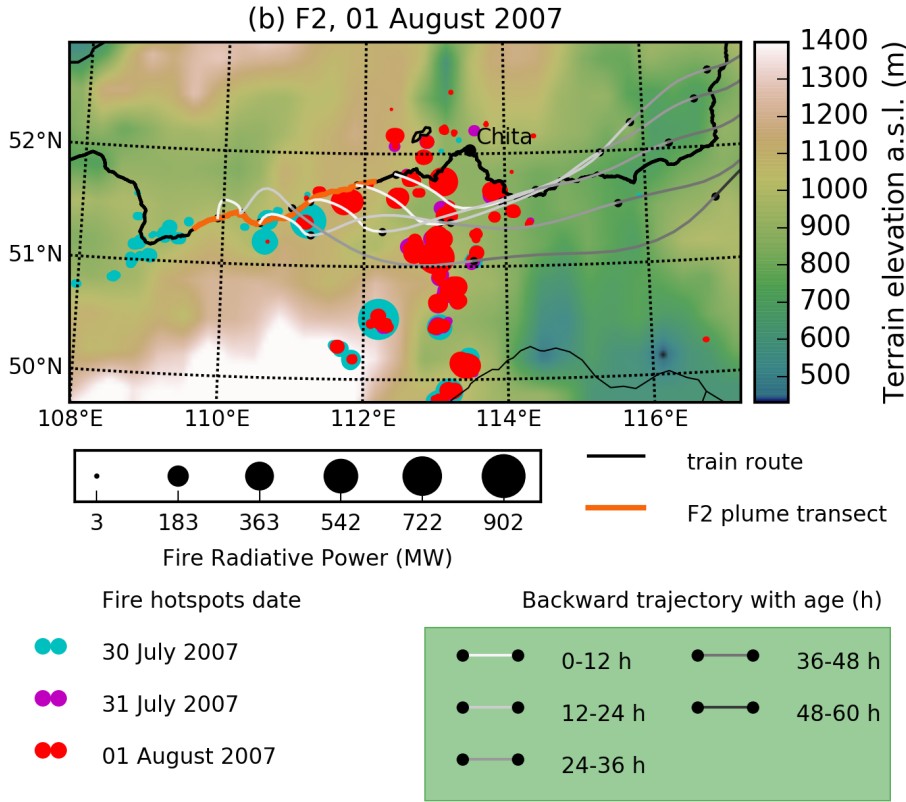

**Figure 2.** Same as Fig. 1 but for the F2 plume. The train course is from East to West.

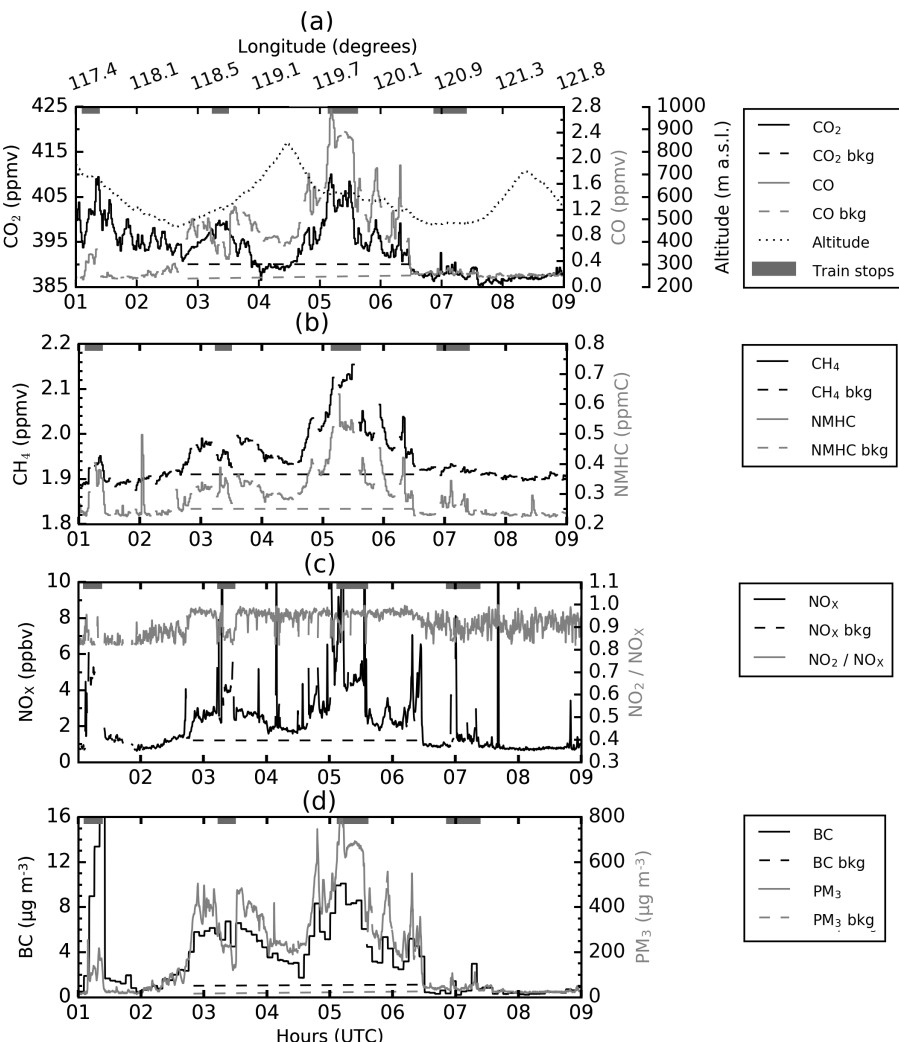

**Figure 3.** Measured 10 s trace gas mixing ratios and aerosol mass concentrations observed in the vicinity of F1 forest fire plume during TROICA-09 expedition on 9 October 2005. Local time is UTC+8.

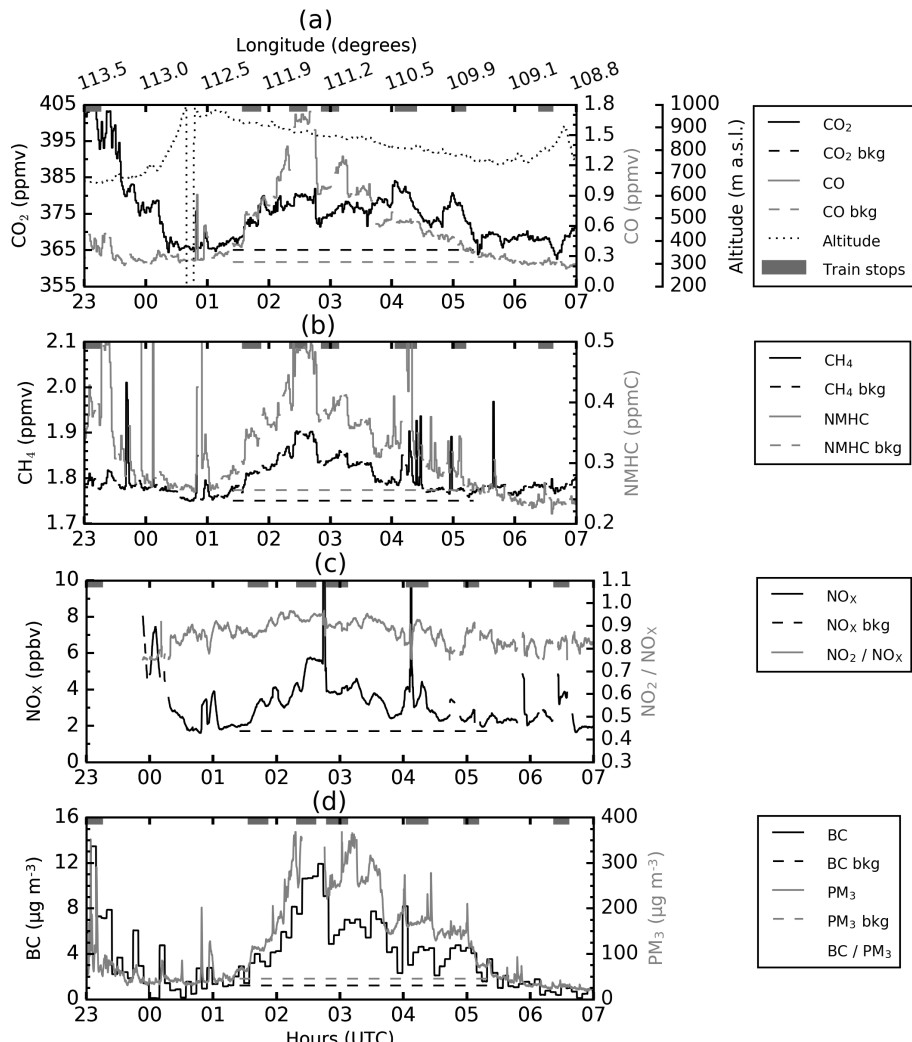

**Figure 4.** Measured 10 s trace gas mixing ratios and aerosol mass concentrations observed in the vicinity of F2 forest fire plume during TROICA-11 expedition on 1 August 2007. Local time is UTC+8.

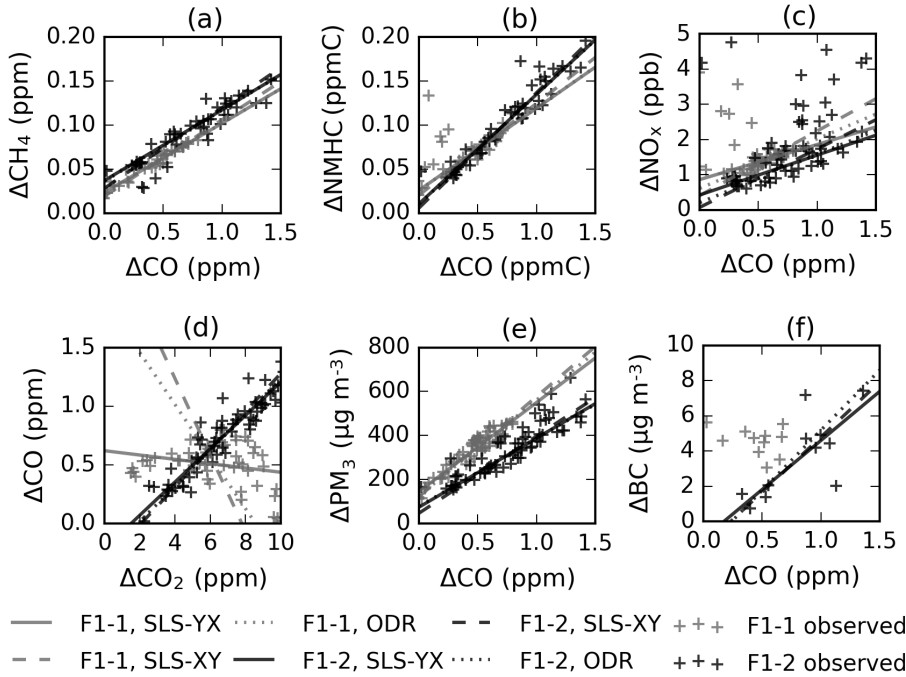

**Figure 5.** Excess levels of trace gases and particles versus excess mixing ratios of $CO$ or $CO_2$ for F1 plume parts, with lines fitted to the data by different regression methods (see explanations in the text).

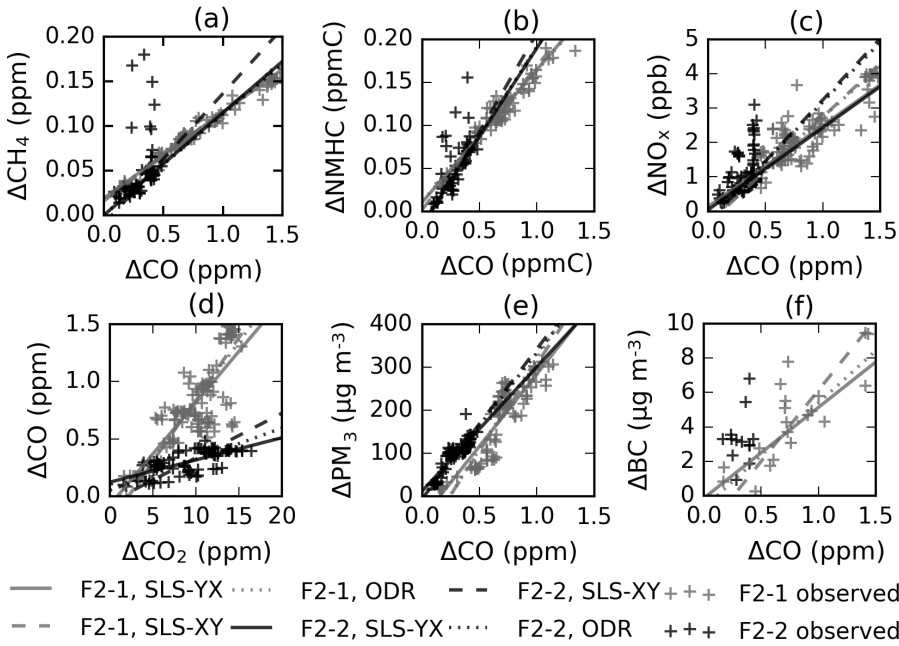

**Figure 6.** Same as Fig. 5 but for F2 plume parts.

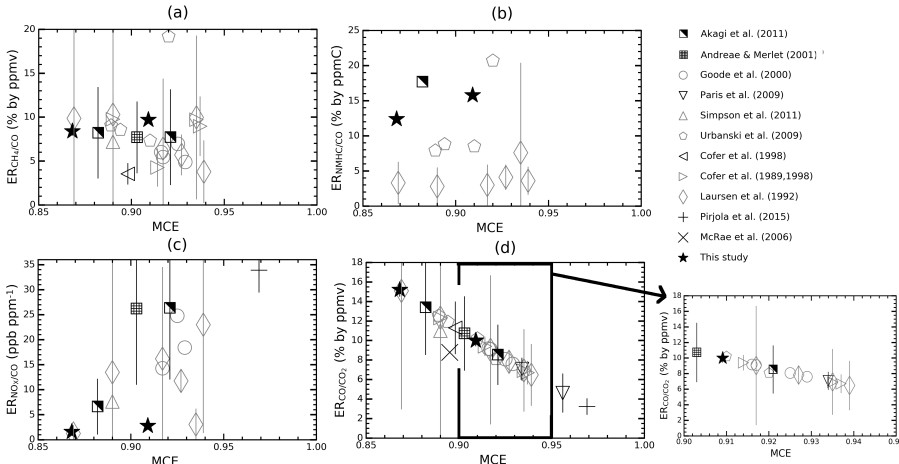

**Figure 7.** Scatter plots of trace gas $ER_{Y/X}$ with standard uncertainties (where available) versus MCE for this study and previous publications.

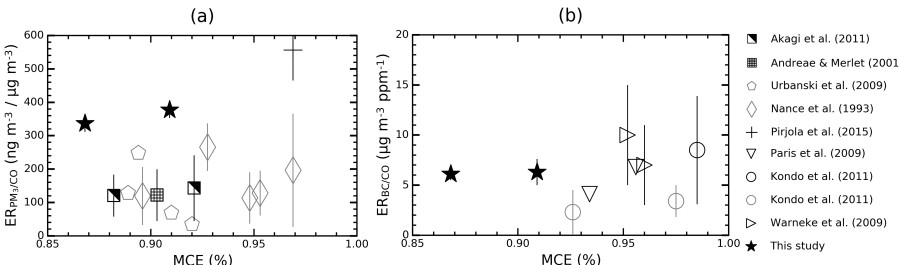

**Figure 8.** Scatter plots of particle $ER_{Y/X}$ with standard uncertainties (where available) versus MCE for this study and previous publications.