# Peer review of "Emission ratios of trace gases and particles for Siberian forest fires on the basis of mobile ground observations"

_Atmospheric Chemistry and Physics, 2017_

## Referee Comment (RC1) · Anonymous Referee #1 · 30 May 2017

Emission ratios of trace gases and particles for Siberian forest fires on the basis of mobile ground observations

Vasileva et al

Review for ACP.

The paper provides a detailed examination of emission ratios of typical gases and aerosol information from two different fire complexes measured from a train in 2005-2007. This is a valuable contribution to constraining the variability and average emission ratios of species from biomass burning. The analysis was carefully thought out, and on the whole the paper is in good shape. However, I do have some concern about

instrumental issues highly relevant to the observations (specifically for the aerosol and BC data), and some suggestions for slightly modifying the analysis/presentation. These are outlined below.

Primary comments:

1) A serious problem is that there is insufficient information about the uncertainties of the measurements – especially the aerosol and black carbon measurements. Essentially all background information about instruments and measurement uncertainty are only referenced via citations, and the citations are insufficient to demonstrate the level of control over the measurements necessary to provide convincing evidence of its value (both because they are hard to find, and because the ones I found did not have enough information). a. For example, Kopeikin 2008 was cited for the GRIMM PM measurements. The entirety of its discussion was: "The nephelometer "Dust Indicator and Tunnel System" designed by GRIMM Corporation (Germany) with the concentration measurement range from 0.01 to 15 ug/mˆ3 was used in expeditions in 2004-2007."

Then there is some discussion of a different system (with limited calibration information) for different missions that are not relevant here. This is entirely inadequate. For BC, I was not able to get the corresponding Kopeikin paper via inter-library-loan, but I saw that it did not contain any references that were relevant to major corrections and uncertainties typically applied and associated with Aethalometer measurements. The current ACPD paper: Comparison of different Aethalometer correction schemes and a reference multi-wavelength absorption technique for ambient aerosol data, by Jorge Saturno et al., gives a good introduction to these issues, which must be dealt with before the data can be considered final. 2) The focus of the paper is on coarsely segregated averages of the two plumes measured, with variability within each section treated more as a "uncertainty source" than as an important feature in its own right. I suggest that the authors attempt to share more information about the variability, perhaps merely via slightly increased discussion, and highlight that component of the results as valuable in their own right to expand the value of these within the whole. In

this vein, I wonder if Figures 3 and 4 could be made more useful by showing the ratios, in addition or instead of the simple concentrations. 3) The connection between the observations of fire state by the scientist on the train, and the actual (mixed) state of the fires at the positions and times actually sourcing the pollution sampled is extremely weak. I did not understand how the line-of-sight of the observer were relevant to large and wide-spread fires. Hence, unless this can be strengthened, I suggest removing the conclusions about lack of connection between flame state and MCE. 4) I did not understand the rationale to omit the CO2 data from a large segment of the F1 plume. How do the authors know that this is not some variability from the fire? Why would CO2 be differently mixed than other trace gases? Unless this is more clearly supported, it seems CO2 should be included for this segment.

Secondary comments:

1) The work of the authors in analyzing the relationships between different species using different fitting techniques was welcome to see. However, the results (and supporting literature) support the idea that the orthogonal distance regression is both the most appropriate approach, and pretty much represented the mean between the other two linear regressions. Once this was clear, I thought the paper might benefit from this discussion being moved to supplemental material, and the graphs simplified by omission of the two simple linear regression lines. After all, this is not the main thrust of the paper. 2) The authors present PM/CO correlations in units of ng mˆ-3/$\mu$g mˆ-3 "for easy comparison to other studies". However, this makes it difficult to compare the other ERs and PM3 to CO correlations, and makes PM/CO a "odd man out". Perhaps it is clearer to leave PM/CO in ng mˆ-3/ppm, and adjust the other studies. 3) In figures 8 and 9 I suggest referring to the new analysis as "this work" rather than Vasileva et al., 2017. 4) The paper is clear and well written, but has numerous small English errors (mostly missing "the"s or extra "thes". Here are some examples: a. Page 2 line 3: "in THE future" b. Line 13: "alter the OXIDATIVE capacity ..." c. Line 14: "and disturb THE background chem..." d. Line 27: "... can SERIOUSLY DETERIORATE the .. "

[Figure]

e. Line 28: "... and contribute to Arctic haze" ("the" should be left out). These, and others, should be corrected.

---

## Referee Comment (RC2) · Anonymous Referee #2 · 7 Jun 2017

Review of acp-2017-362

Emission ratios of trace gases and particles for Siberian forest fires on the basis of mobile ground observations

General Comments

This paper reports emission ratios for Siberian forest fires measured from a mobile ground based platform. The measurements were obtained from a laboratory car on an electric powered train travelling the Trans-Siberian Railway in October 2005 and August 2007. In each year the train passed through smoke plumes from nearby forest fires. Each smoke plume crossing event was ïA¿200 km in length and ïA¿ 4 hours in

duration. Instrumentation on the lab car measured CO2, CO, CH4, NMHC, NO, NO2, PM3, and BC. The study provides emission ratios which are an important addition to the limited body of knowledge regarding emissions from boreal forest fires in northern Eurasia. Given the global importance of northern Eurasia as a source of biomass burning emissions and sensitivity the sensitivity of the region to climate and associated likelihood of increased fire in the feature, this study presents a potentially important contribution to the biomass burning science literature. However, besides serval specific comments, I have found three issues that must be addressed prior to publication:

Age and source of smoke

This paper reports observations of normalized excess mixing ratios (NEMR) which they classify as emission ratios (ER) based on the assertion that the smoke plumes sampled were less than 24 hours old. A NEMR is only an ER if the smoke has not undergone significant chemical transformation. ER may be used to derive emission factors (EF) for estimating mass emissions when combined with estimated of fuel mass consumed. Often 24 hours is used as an arbitrary threshold for classifying NEMR as ER (see below). However, the authors have not provided evidence demonstrating that the plumes sampled were less than 24 hours old. Figure 2 maps back trajectories, plume transects, and CO emissions totaled over a two month period. Figure 2 provides no insight into where fires were active during the day of sampling or the preceding few days which may have contributed to the emissions measured. The authors need to provide a better demonstration of the rough plume age. For example map MODIS active fire detections for the day of and preceding few days of the plume samples. Use larger figures with focused on the area of interest with back trajectories labeled for time. I suggest something similar to the presentation in the supplementary material of (Collier et al., 2016). With only two samples periods (2 plumes) this should not be difficult to do. In its current state, the paper doesn't demonstrate the approximate plume age or reasonably identify the source regions; therefore the assertion that the smoke samples may be used as ER is cannot be accepted.

[Figure]

Normalized excess mixing ratios, emission ratios, and uncertainties

The Methods section needs a more complete description of emission measurements along the line alluded to at P12 L33-P13, L2. The authors need to distinguish between excess mixing ratios, normalized excess mixing ratios, and the conditions under which a normalized excess mixing ratio may be considered an emission ratio (ER). A few points (Akagi et al., 2011; Yokelson et al., 2013): The excess mixing ratio of species X in a plume is dX = dXplume – dXbackground. The normalized excess mixing ratio (NEMR) is dX/dY, where Y is a long-lived reference species co-emitted with X, CO or $CO_2$, to normalize for dilution (Equation 1 in manuscript). If "fresh emissions" are measured, then the NEMR is an "emission ratio" (ER) which can be used to derive emission factors (EF) which may be used to estimate emissions per unit mass of fuel consumed. To be characterized as fresh emissions there must be no significant photochemical loss or other removal or production of either X or Y (Yokelson et al., 2013). Assigning a simple age since emission as a threshold for when a NEMR may be considered an ER that can be used to derive EF involves much uncertainty. The destruction or creation of an emitted species X depends on a host of factors including the chemical reactivity, volatility, and photolability of X, the composition of the emissions, the plume dilution rate and dispersion conditions, composition of the background air that mixes with the plume, and solar insolation.

Additionally, it should be noted for readers that field measurements from aircraft platforms have observed changes in smoke plume chemical composition within 0.5 to 5 hours after emissions (Akagi et al., 2013, 2012; Liu et al., 2016; May et al., 2015) I do not argue that smoke which is one day old cannot be used to report ER. The "one day" threshold, while somewhat arbitrary, has been widely used (Hornbrook et al., 2011; O'Shea et al., 2013; Simpson et al., 2011). However, it is important that readers that when smoke is not sampled at the source there are significant uncertainties when using these smoke samples to assign ER and/or EF.

Treatment of observations

P10, L3-7: "The observed strong scattering of some data subsets is clearly attributable to highly complex measurement environment and the supposed strong spatial heterogeneity of the emission sources contributed to the smoke plumes. Consequently, we exclude from the analyzes the measurements producing extremely high or low dY/dX values to make our final estimates more robust with respect to various disturbing factors."

This is not an appropriate manner to handle the data. One cannot simply toss data points because they introduce scatter and reduce the correlation coefficient and increase the uncertainty of the slope in the assumed relationship. The authors should have an objective criteria for identifying data segments that are treated as the biomass smoke plume. Rejection of observations taken within the biomass plume should only be rejected using a clear, objective criteria that is based on sound reasoning – e.g. a significant influence of a local anthropogenic, instrument malfunction, or failed calibration.

Specific Comments

P3, L23-24: "Both the plumes were observed in Transbaikalia – a mountainous area in the south Siberia east to the Lake Baikal known for its severe wildfire activity during warm seasons which start early in spring due to exceptionally dry weather conditions" This sentence is awkward and I do not understand the last portion.

Measurements and instrumentation

Grimm calibration PM3 was measured by light scattering which depends in part on the particle size distribution, chemical composition, and morphology. Please clarify if the PM3 mass density reported is based on the instrument's factory calibration or if it was calibrated for biomass burning aerosols(Aurell and Gullett, 2013; Yokelson et al., 2007) and (Nance et al., 1993). If the instrument's factory calibration was used do you anticipate any systemic bias for biomass smoke aerosols?

NMHC detection of OVOC In biomass smoke a significant fraction of VOC are oxygenated-VOC (OVOC) (Akagi et al., 2011; Gilman et al., 2015). Please comment on the sensitivity of the study's NMHC detection method to OVOC, in particular the possible under-sampling of these compounds, e.g. Trabue et al., 2013.

Results and discussion Please describe how the smoke plume boundaries were identified /selected. Were they selected based on PM3 level, coincident increases in PM3 and CO, or some other criteria?

How did the authors assign observations to the different plume segments? Do the plume segments, e.g. F1-1 and F1-2, correspond to different stretches of the sample path? Please clarify. The different plume segments need to be identified on Figures 3 & 4.

P9, L9-11: Simpson et al. (2011) data show $dNO2/dNOx \sim 70\%$.

P9, L13-17 and Figs. 3 & 4 Do the "train stops" regions highlighted at the top of the plots correspond to regions excluded from the analysis?

P9, L15-17: Please explain how/why these criteria for identifying anthropogenic contamination were selected.

Figure 3 & 4. Do the dashed background lines correspond to the plume sample period? Please clarify. Figures 3 & 4 should be plotted with local time or note the offset in the caption.

P9, L19: I assume "500 to 800 m a.g.l." should be "500 to 800 m a.s.l." i.e. meters above sea-level.

Tables 5 & 6 should be merged.

P10, L1-3: NOx and BC are associated with flaming combustion and may correlate better with $CO2$. Did the authors check for correlation vs. $CO2$ and if so how does it compare with that vs. CO?

P10, L3-7: "The observed strong scattering of some data subsets is clearly attributable to highly complex measurement environment and the supposed strong spatial heterogeneity of the emission sources contributed to the smoke plumes. Consequently, we exclude from the analyzes the measurements producing extremely high or low dY/dX values to make our final estimates more robust with respect to various disturbing factors."

This is not an appropriate manner to handle the data. One cannot simply toss data points because they introduce scatter and reduce the correlation coefficient and increase the uncertainty of the slope in the assumed relationship. The authors should have an objective criteria for identifying data segments that are treated as the biomass smoke plume. Rejection of observations taken within the biomass plume should only be rejected using a clear, objective criteria that is based on sound reasoning – e.g. a significant influence of a local anthropogenic, instrument malfunction, or failed calibration.

Also, it is unclear what is meant by: "...more robust with respect to various disturbing factors"

P10, L8-16: I suspect a portion of the plume F2-2 was influenced by a biogenic $CO_2$ source. Examination of Fig 6d and Fig 4a leads me to believe that F2-2 corresponds to the second portion of the plume around 3:30 to 5:30 UTC, which exhibits to broad peaks in $CO_2$ between 4:00 and 5:30 UTC for which there is not coinciding response in the CO. Additionally, the NOx does not show not increase during these broad $CO_2$ peaks (Figure 4c). Since NOx is associated with flaming combustion one would expect it to correlate with $CO_2$. Since it does not, this is further evidence that the $CO_2$ mixing ratio sampled during this plume stretch is noticeably influenced by a non-fire source. Also, the dNOx/dCO ratio for F2-1 and F2-2 are the same within uncertainties ($2.8\pm0.2$ versus $3.1\pm0.4$). If the source of plume segments F2-1 and F2-2 was really a fires with MCE of 0.91 and 0.97, respectively, one would expect a difference in dNOx/dCO. I strongly disagree with the authors' interpretation of Figure 7b. It appears

that dBC/dPM3 are very similar for F2-1 and F2-2. What are the plume segment average values for these ratios? I find it difficult to believe they are significantly different. In fact, I interpret Fig 7b as evidence that segments F2-1 and F2-2 originated from fires with very similar MCE. The authors should consider the CO2 during this stretch to be highly suspect and not report dCO/dCO2 or MCE for this segment.

P10, L32 – P11, L2: Based on my comments, I do not believe F2-2 should be considered flaming. I would limit comparison to F1-2 and F2-1, since these have valid MCE.

P11, L14-16: The authors have not demonstrated the sampled plumes are likely less than 1 day old (see general comments).

P13, Ln 27-29: I believe these were not included in Akagi et al. (2011) as they did not measure "fresh smoke" samples or the smoke age was uncertain.

P16, Ln32-33 While the authors report the train operator observed some fire activity, they are clear in stating that the plumes sampled likely resulted from multiple fires, all of which were not observed. Therefore, the authors cannot relate their measured MCE to any specific observed combustion type. I agree that visual observations of fire behavior tend to be a poor metric for classifying combustion type and MCE, especially since both flaming and smoldering typically occur simultaneously for naturally burning forest fires. However, given that EF for many species are correlated with MCE, it does have utility for extrapolating measured EF to other fire types with different MCE regimes.

Comparison with other published results The discussion and figures are a bit confusing. The authors seem to include studies where the plumes sampled were older than 1 day and therefore are not emission ratios and not appropriate for comparison with the current work. I strongly recommend the authors limit the comparison to studies where the plume samples were <= 1 day old and result from boreal fires.

NMHC comparison and Figure 8b: The NMHC EF based on Laursen et al. (1992) and

Urbanski et al. (2009) are the sum of only a handful of compounds and not comprehensive VOC measurement like that constructed in the current study. This should be clarified in the text.

Technical Corrections The authors should define chemical formulas when first introduced. P1, L16: Insert "the" between (btw) "and" and "boreal" P1, L16: change "became" to "become" P1, L18: insert "the" btw "including" and "global" P2, L3: Insert "the" btw "In" and "future" P2, L14: Change "OH-" to "OH" it's a radical not an ion. No charge. P2, L17: change "is" to "are" P2, L19: change "on the basis" to "by" P2, L21: delete ",Canada, and Alaska as a . . .." P3, L1: insert "of' btw "all" and "these" P3, L17: change "substantia amount" to "many" P3, L32: insert "that" before "originated"

There are many similar errors in English usage throughout the remainder of the manuscript that need correction.

Figure 3a – The CO and CO2 background lines have wrong colors

Table 6 change "PM1" to "PM3"

Figures 8 & 9: The plotted symbols do not all match the legend, Vasileva et al., 2017 and Pirjola et al., 2015 are different.

References Steven Trabue , Kenwood Scoggin , Laura L. McConnell , Hong Li , Andrew Turner , Robert Burns , Hongwei Xin , Richard S. Gates , Alam Hasson , Segun Ogunjemiyo , Ronaldo Maghirang & Jerry Hatfield (2013) Performance of commercial non-methane hydrocarbon analyzers in monitoring oxygenated volatile organic compounds emitted from animal feeding operations, Journal of the Air & Waste Management Association, 63:10, 1163-1172, DOI:10.1080/10962247.2013.804464

Akagi, S.K., Craven, J.S., Taylor, J.W., McMeeking, G.R., Yokelson, R.J., Burling, I.R., Urbanski, S.P., Wold, C.E., Seinfeld, J.H., Coe, H., Alvarado, M.J., Weise, D.R., 2012. Evolution of trace gases and particles emitted by a chaparral fire in California. Atmos Chem Phys 12, 1397–1421. doi:10.5194/acp-12-1397-2012
Akagi, S.K., Yokelson, R.J., Burling, I.R., Meinardi, S., Simpson, I., Blake, D.R., McMeeking, G.R., Sullivan, A., Lee, T., Kreidenweis, S., Urbanski, S., Reardon, J., Griffith, D.W.T., Johnson, T.J., Weise, D.R., 2013. Measurements of reactive trace gases and variable O3 formation rates in some South Carolina biomass burning plumes. Atmos Chem Phys 13, 1141–1165. doi:10.5194/acp-13-1141-2013

Akagi, S.K., Yokelson, R.J., Wiedinmyer, C., Alvarado, M.J., Reid, J.S., Karl, T., Crounse, J.D., Wennberg, P.O., 2011. Emission factors for open and domestic biomass burning for use in atmospheric models. Atmos Chem Phys 11, 4039–4072. doi:10.5194/acp-11-4039-2011

Aurell, J., Gullett, B.K., 2013. Emission Factors from Aerial and Ground Measurements of Field and Laboratory Forest Burns in the Southeastern U.S.: PM2.5, Black and Brown Carbon, VOC, and PCDD/PCDF. Environ. Sci. Technol. 47, 8443–8452. doi:10.1021/es402101k

Collier, S., Zhou, S., Onasch, T.B., Jaffe, D.A., Kleinman, L., Sedlacek, A.J., Briggs, N.L., Hee, J., Fortner, E., Shilling, J.E., Worsnop, D., Yokelson, R.J., Parworth, C., Ge, X., Xu, J., Butterfield, Z., Chand, D., Dubey, M.K., Pekour, M.S., Springston, S., Zhang, Q., 2016. Regional Influence of Aerosol Emissions from Wildfires Driven by Combustion Efficiency: Insights from the BBOP Campaign. Environ. Sci. Technol. 50, 8613–8622. doi:10.1021/acs.est.6b01617

Gilman, J.B., Lerner, B.M., Kuster, W.C., Goldan, P.D., Warneke, C., Veres, P.R., Roberts, J.M., de Gouw, J.A., Burling, I.R., Yokelson, R.J., 2015. Biomass burning emissions and potential air quality impacts of volatile organic compounds and other trace gases from fuels common in the US. Atmos Chem Phys 15, 13915–13938. doi:10.5194/acp-15-13915-2015

Hornbrook, R.S., Blake, D.R., Diskin, G.S., Fried, A., Fuelberg, H.E., Meinardi, S., Mikoviny, T., Richter, D., Sachse, G.W., Vay, S.A., Walega, J., Weibring, P., Weinheimer, A.J., Wiedinmyer, C., Wisthaler, A., Hills, A., Riemer, D.D., Apel, E.C., 2011.

Observations of nonmethane organic compounds during ARCTAS − Part 1: Biomass burning emissions and plume enhancements. Atmos Chem Phys 11, 11103–11130. doi:10.5194/acp-11-11103-2011

Liu, X., Zhang, Y., Huey, L.G., Yokelson, R.J., Wang, Y., Jimenez, J.L., Campuzano-Jost, P., Beyersdorf, A.J., Blake, D.R., Choi, Y., St. Clair, J.M., Crounse, J.D., Day, D.A., Diskin, G.S., Fried, A., Hall, S.R., Hanisco, T.F., King, L.E., Meinardi, S., Mikoviny, T., Palm, B.B., Peischl, J., Perring, A.E., Pollack, I.B., Ryerson, T.B., Sachse, G., Schwarz, J.P., Simpson, I.J., Tanner, D.J., Thornhill, K.L., Ullmann, K., Weber, R.J., Wennberg, P.O., Wisthaler, A., Wolfe, G.M., Ziemba, L.D., 2016. Agricultural fires in the southeastern U.S. during SEAC 4 RS: Emissions of trace gases and particles and evolution of ozone, reactive nitrogen, and organic aerosol: Agricultural Fires in the SE US. J. Geophys. Res. Atmospheres 121, 7383–7414. doi:10.1002/2016JD025040

May, A.A., Lee, T., McMeeking, G.R., Akagi, S., Sullivan, A.P., Urbanski, S., Yokelson, R.J., Kreidenweis, S.M., 2015. Observations and analysis of organic aerosol evolution in some prescribed fire smoke plumes. Atmos Chem Phys 15, 6323–6335. doi:10.5194/acp-15-6323-2015

O'Shea, S.J., Allen, G., Gallagher, M.W., Bauguitte, S.J.-B., Illingworth, S.M., Le Breton, M., Muller, J.B.A., Percival, C.J., Archibald, A.T., Oram, D.E., Parrington, M., Palmer, P.I., Lewis, A.C., 2013. Airborne observations of trace gases over boreal Canada during BORTAS: campaign climatology, air mass analysis and enhancement ratios. Atmos Chem Phys 13, 12451–12467. doi:10.5194/acp-13-12451-2013

Simpson, I.J., Akagi, S.K., Barletta, B., Blake, N.J., Choi, Y., Diskin, G.S., Fried, A., Fuelberg, H.E., Meinardi, S., Rowland, F.S., Vay, S.A., Weinheimer, A.J., Wennberg, P.O., Wiebring, P., Wisthaler, A., Yang, M., Yokelson, R.J., Blake, D.R., 2011. Boreal forest fire emissions in fresh Canadian smoke plumes: $C_1$-$C_{10}$ volatile organic compounds (VOCs), $CO_2$, CO, $NO_2$, NO,

HCN and CH$_3$CN. Atmospheric Chem. Phys. 11, 6445–6463. doi:10.5194/acp-11-6445-2011

Yokelson, R.J., Andreae, M.O., Akagi, S.K., 2013. Pitfalls with the use of enhancement ratios or normalized excess mixing ratios measured in plumes to characterize pollution sources and aging. Atmos Meas Tech 6, 2155–2158. doi:10.5194/amt-6-2155-2013

Yokelson, R.J., Urbanski, S.P., Atlas, E.L., Toohey, D.W., Alvarado, E.C., Crounse, J.D., Wennberg, P.O., Fisher, M.E., Wold, C.E., Campos, T.L., Adachi, K., Buseck, P.R., Hao, W.M., 2007. Emissions from forest fires near Mexico City. Atmos Chem Phys 7, 5569–5584. doi:10.5194/acp-7-5569-2007

---

## Author Comment (AC1) · 4 Aug 2017

The authors thank the Referee 1 for a favourable attention to the manuscript as well as for useful comments which helped us to improve presentation of the results.

**1  Primary comments**

**Comment.**  1) A serious problem is that there is insufficient information about the uncertainties of the measurements – especially the aerosol and black carbon mea-

surements. Essentially all background information about instruments and measurement uncertainty are only referenced via citations, and the citations are insufficient to demonstrate the level of control over the measurements necessary to provide convincing evidence of its value (both because they are hard to find, and because the ones I found did not have enough information). a. For example, Kopeikin 2008 was cited for the GRIMM PM measurements. The entirety of its discussion was: "The nephelometer "Dust Indicator and Tunnel System" designed by GRIMM Corporation (Germany) with the concentration measurement range from 0.01 to 15 $\mathrm{mg\ m^{-3}}$ was used in expeditions in 2004-2007." Then there is some discussion of a different system (with limited calibration information) for different missions that are not relevant here. This is entirely inadequate. For BC, I was not able to get the corresponding Kopeikin paper via interlibrary-loan, but I saw that it did not contain any references that were relevant to major corrections and uncertainties typically applied and associated with Aethalometer measurements. The current ACPD paper: Comparison of different Aethalometer correction schemes and a reference multi-wavelength absorption technique for ambient aerosol data, by Jorge Saturno et al., gives a good introduction to these issues, which must be dealt with before the data can be considered final.

**Response.** We have essentially expanded the "Measurements ans instrumentation" section to address your comment. Here we may add that natural variability of wildfire emissions is significant and cause variability in the measured concentrations (see Fig. 3–4) that is much higher than potential instrument errors. Anyway, the high correlations between $\Delta PM_3$ and $\Delta CO$ reported in the study suggest that we should not expect significant errors due to the PM3 measurements technique in the estimated fire plume $\Delta PM_3/\Delta CO$ ratios. Yes, a sort of a systematic bias in the PM3 vs. CO scatter plots seen in Fig. 5e and Fig. 6e may be caused by either the lack of calibration or by natural variations of the emissions as well. Therefore, we suggest using the plume average $ER_{PM3/CO}$ in the revised manuscript to compensate this potential issue. Same is true for the BC measurements. As we can see from the relevant papers, the corrections of BC measurements by multi-wave method, as well as the correction of

nephelometer calibration with respect to particle size distribution, chemical composition and morphology in biomass smoke may be within several percents that is considerably lower than variability of the measurements caused by natural factors (changies in intensity of burning, flaming vs. smoldering combustion, dilution during dispersion in the atmosphere) that are addressed during the analysis. Note also the relatively low temporal resolution (5 min) of the BC measurements contributing to high (20%) uncertainties in the $ER_{BC/CO}$ estimates as well. To our opinion, the principal value of the TROICA measurements is their transcontinental extent and uniqueness because the TROICA routes covered the regions for which very little observational data is available today. This somewhat compensates the fact that the TROICA's measurements may be not as precise as the up-to-date ones.

**Changes.** The "Measurements ans instrumentation" is essentially expanded to provide information on the measurements instrument calibration. See also in the "Results and discussion" section:

"The estimated $ER_{PM3/CO}$ varies within 320–385 $\frac{\text{ng m}^{-3}}{\text{µg m}^{-3}}$ with the relative uncertainties of 4–8% caused mainly by variability in the measured concentrations which, in turn, may come either from natural variability of fire emissions or from aerosol specific measurement errors. The latter are most probably related to the specific features of biomass smoke aerosol which were not completely accounted for during the instrument calibration as pointed above."

"...the $ER_{PM3/CO}$ which varies by 50–55 $\frac{\text{ng m}^{-3}}{\text{µg m}^{-3}}$ within each plume. The latter may be due to the incomplete calibration of the $PM_3$ measurement instrument for biomass smoke aerosol as pointed above, therefore we may suggest to use the average $ER_{PM3/CO}$ for each plume (which is about $360 \pm 30$ $\frac{\text{ng m}^{-3}}{\text{µg m}^{-3}}$ for F1 and $350 \pm 32$ $\frac{\text{ng m}^{-3}}{\text{µg m}^{-3}}$ for F2) to address this issue."

**Comment.** 2) The focus of the paper is on coarsely segregated averages of the two plumes measured, with variability within each section treated more as a "uncertainty

source" than as an important feature in its own right. I suggest that the authors attempt to share more information about the variability, perhaps merely via slightly increased discussion, and highlight that component of the results as valuable in their own right to expand the value of these within the whole. In this vein, I wonder if Figures 3 and 4 could be made more useful by showing the ratios, in addition or instead of the simple concentrations.

**Response.** Thank you for this very important comment. Although the physical causes (including natural variability) of uncertainty for each ER estimate are considered in the manuscript, the topic needs additional discussion. Thus we've extended the discussion of the variability and revised the interpretation: indeed, variability of gas ERs within the plumes is small (the exceptions are discussed), while variability between the plumes is noticeable. We've also revised the interpretation of variations in the estimated ERs with respect to the source-receptor relationships for the F1 and F2 plumes (to address the Referee 2 comments) and concluded that the effect of physical and chemical transformations related to plume ageing on the ER variability was not significant. For details please refer to the changes in the manuscript listed below.

We have also considered showing the time series of emission ratios in Fig. 3-4 but addition of new lines makes plots difficult to read. At the same time, we'd really like to show the original measurements data and the observed magnitudes of real concentrations, because it is the base for all the research. Therefore, we have decided to show the original data in Fig.3-4, and the scattering of the emission ratios in Fig. 5-6. We would leave Fig. 3-4 as they are, if you don't mind.

**Changes.** Additional potential causes of the variability (uncertainty) in individual ERs are pointed at the end of the corresponding paragraphs in "Results and Discussion" section. We've also added three paragraphs at the end of the "Results and Discussion" section with an additional discussion of the ER variability within and between different plume segments, as well as between different plumes:

"In the following paragraphs we summarize the uncertainty and variability in the ER estimates reported in Table 5. In the individual ER estimates the ranges of relative variations..."

"The variability of the reported $ER_{avg}$ between different plume segments within each plume generally does not exceed the variability..."

"We note finally, that the variability of ERs between F1 and F2 plumes is more pronounced than within each plume..."

See also:

"Herewith the term "uncertainty" means the precision of a model estimate as well as natural variability of the estimated quantity, because both these meanings are closely related in the present study."

In the "Conclusions" section see the following paragraphs:

"Between the plumes, the estimated gas ERs vary appreciably due to..."

"The uncertainties in the ER estimates are associated mainly with..."

Also see:

"The derived gas ERs are generally stable within the plumes, with the differences between the ERs estimated for different plume segments being statistically insignificant, which supports the general idea of a common fire smoke age throughout each plume, as well as a negligible effect of the changing environment on the measurements."

**Comment.** 3) The connection between the observations of fire state by the scientist on the train, and the actual (mixed) state of the fires at the positions and times actually sourcing the pollution sampled is extremely weak. I did not understand how the line-of-sight of the observer were relevant to large and wide-spread fires. Hence, unless this can be strengthened, I suggest removing the conclusions about lack of connection between flame state and MCE.

**Response.** The remark is rather true for the present study, although the source-receptor relationships were revised in the manuscript following the critical comments of the Referee2. Anyway, the authors cite previous publications where atypical combination of MCE and visual observations were reported (see page 12, lines 17-24 of the discussed non-revised manuscript) and yield confusing results with high MCE attributed to smoldering (Pirjola et al., 2015) and low MCE attributed to flaming (Cofer et al., 1998). To clarify this issue, we have changed this pat of the "Conclusions" section as follows.

**Changes.** In the "Conclusions" section:

"The authors did not find any definite relation between the visually observed combustion type (smoldering or flaming) and MCE values neither in this study, mainly because of the lack of detailed information on fire state, nor in the previous studies where emissions from experimental fires were attributed to flaming or smoldering combustion on the basis of visual inspections (Cofer et al., 1998; Pirjola et al., 2015). Thus, we are cautious in using visual observations to attribute fire emissions to a specific combustion type since both flaming and smoldering typically occur simultaneously for naturally burning forest fires."

See also the revised source-receptor relationships in the "Plume crossing episodes" section (discussion of Fig. 1–2) which has been substantially changed following critical comments of the Referee 2. In the "Results and discussion" section, see the TWO paragraphs starting at "Beyond the data segments corresponding to the train stops described above, peak excess levels within both the plumes were observed near the locations of active fires detected exactly in the day of the plume observation directly close to the railway..."

**Comment.** 4) I did not understand the rationale to omit the CO2 data from a large segment of the F1 plume. How do the authors know that this is not some variability from the fire? Why would $CO_2$ be differently mixed than other trace gases?

**Response.** We suspect contribution from a non-fire $CO_2$ emissions during 02:50–04:35 UTC in F1 plume and 03:40–05:20 UTC in F2 plume, with the latter suggested by the Referee 2. Therefore we do not report $ER_{CO/CO2}$ and MCE for those plume parts (F1-1 and F2-2). Please see details in the revised manuscript.

**Changes.** Two paragraphs are added into the "Results and discussion" section to describe episodes of possible contamination by non-fire emissions within the plumes:

"Thus, Fig. 3 shows a distinct decrease in all excess mixing ratios..."

"In Fig. 4a two broad $CO_2$ peaks in the western F2 plume part..."

**2   Secondary comments**

**Comment.** 1) The work of the authors in analyzing the relationships between different species using different fitting techniques was welcome to see. However, the results (and supporting literature) support the idea that the orthogonal distance regression is both the most appropriate approach, and pretty much represented the mean between the other two linear regressions. Once this was clear, I thought the paper might benefit from this discussion being moved to supplemental material, and the graphs simplified by omission of the two simple linear regression lines. After all, this is not the main thrust of the paper.

**Response.** Thank you for an interesting suggestion. However, we can not say that the orthogonal distance regression (ORD) method is the most appropriate (see P7, L22–25 in the discussion manuscript). If you look at Fig. 5-6 you'll see that the ODR line also does not lay between the other two regression lines in the most controversial cases (Fig. 6a,c,e,f, Fig. 5f). Further research of this topic refers to the study of mathematical methods which is interesting but falls beyond the scope of the discussed manuscript. Hence we had to keep the three lines at the graphs to enable the skeptical readers to

visually assess the representativeness of the average ERs listed in Table 5.

**Comment.** 2) The authors present PM/CO correlations in units of $(\text{ng m}^{-3})/(\text{µg m}^{-3})$ "for easy comparison to other studies". However, this makes it difficult to compare the other ERs and $PM_3$ to CO correlations, and makes $PM_3$/CO a "odd man out". Perhaps it is clearer to leave $PM_3$/CO in $\text{ng m}^{-3} \text{ ppm}^{-1}$, and adjust the other studies.

**Response.** It is reasonable to do that, but the $PM_3$ emissions in the cited studies are expressed as emission factors $(\text{g kg}^{-1})$, thus calculation of $ER_{PM3/CO}$ in units of $\text{ng m}^{-3} \text{ ppm}^{-1}$ requires information about atmospheric pressure and temperature which was not available in all the cited studies and therefore introduces additional uncertainty. In our study, the $ER_{PM3/CO}$ in units of $\text{ng m}^{-3} \text{ ppm}^{-1}$ seemed to depend on the changing altitude in the F1 plume, therefore we chose the unit of $(\text{ng m}^{-3})/(\text{µg m}^{-3})$ as the more robust one.

**Comment.** 3) In figures 8 and 9 I suggest referring to the new analysis as "this work" rather than Vasileva et al., 2017.

**Response.** Thanks. Done.

**Comment.** 4) The paper is clear and well written, but has numerous small English errors (mostly missing "the"s or extra "thes". Here are some examples:

a. Page 2 line 3: "in THE future"

b. Line 13: "alter the OXIDATIVE capacity"

c. Line 14: "and disturb THE background chem"

d. Line 27: "... can SERIOUSLY DETERIORATE the .. "

e. Line 28: "... and contribute to Arctic haze" ("the" should be left out)."

These, and others, should be corrected.

**Response.** Many thanks for your attention. We have tried to do our best.

---

## Author Comment (AC3) · 4 Aug 2017

The authors thank the Referee 2 for a careful examination of the the manuscript and a favorable general comment. We also thank for a constructive discussion and valuable suggestions which helped us to clarify presentation of the results. The comment on the plume age forced us to revise the source-receptor relationships for F1 and F2 plumes which affected interpretation of the estimated ER variability. Please find below the answers to all the critical comments and the relevant changes in the manuscript.

[Figure]

**1   General comments**

**Comment: Age and source of smoke.** This paper reports observations of normalized excess mixing ratios (NEMR) which they classify as emission ratios (ER) based on the assertion that the smoke plumes sampled were less than 24 hours old. A NEMR is only an ER if the smoke has not undergone significant chemical transformation. ER may be used to derive emission factors (EF) for estimating mass emissions when combined with estimated of fuel mass consumed. Often 24 hours is used as an arbitrary threshold for classifying NEMR as ER (see below). However, the authors have not provided evidence demonstrating that the plumes sampled were less than 24 hours old. Figure 2 maps back trajectories, plume transects, and $CO$ emissions totaled over a two month period. Figure 2 provides no insight into where fires were active during the day of sampling or the preceding few days which may have contributed to the emissions measured. The authors need to provide a better demonstration of the rough plume age. For example map MODIS active fire detections for the day of and preceding few days of the plume samples. Use larger figures with focused on the area of interest with back trajectories labeled for time. I suggest something similar to the presentation in the supplementary material of (Collier et al., 2016). With only two samples periods (2 plumes) this should not be difficult to do. In its current state, the paper doesn't demonstrate the approximate plume age or reasonably identify the source regions; therefore the assertion that the smoke samples may be used as ER is cannot be accepted.

**Comment: Normalized excess mixing ratios, emission ratios, and uncertainties.** The Methods section needs a more complete description of emission measurements along the line alluded to at P12 L33-P13, L2. The authors need to distinguish between excess mixing ratios, normalized excess mixing ratios, and the conditions under which a normalized excess mixing ratio may be considered an emission ratio (ER). A few points (Akagi et al., 2011; Yokelson et al., 2013): The excess mixing ratio of species X in a plume is dX = dXplume − dXbackground. The normalized excess mixing ratio
(NEMR) is dX/dY, where Y is a long-lived reference species co-emitted with X, CO or CO2, to normalize for dilution (Equation 1 in manuscript). If "fresh emissions" are measured, then the NEMR is an "emission ratio" (ER) which can be used to derive emission factors (EF) which may be used to estimate emissions per unit mass of fuel consumed. To be characterized as fresh emissions there must be no significant photochemical loss or other removal or production of either X or Y (Yokelson et al., 2013). Assigning a simple age since emission as a threshold for when a NEMR may be considered an ER that can be used to derive EF involves much uncertainty. The destruction or creation of an emitted species X depends on a host of factors including the chemical reactivity, volatility, and photolability of X, the composition of the emissions, the plume dilution rate and dispersion conditions, composition of the background air that mixes with the plume, and solar insolation. Additionally, it should be noted for readers that field measurements from aircraft platforms have observed changes in smoke plume chemical composition within 0.5 to 5 hours after emissions (Akagi et al., 2013, 2012; Liu et al., 2016; May et al., 2015) I do not argue that smoke which is one day old cannot be used to report ER. The "one day" threshold, while somewhat arbitrary, has been widely used (Hornbrook et al., 2011; O'Shea et al., 2013; Simpson et al., 2011). However, it is important that readers that when smoke is not sampled at the source there are significant uncertainties when using these smoke samples to assign ER and/or EF.

**Response to the comments.** Thank you for these essential comments. Here we reply to them both. We've edited Fig. 1–2 and Fig. 3–4 to demonstrate the source-receptor relationships for F1 and F2 fire plumes. Now in Fig. 1–2 the MODIS active fires are shown as circles with the size proportional to fire radiative power (FRP) and color indicating the day o fire detection. Possible origins of the air sampled within the plumes are shown in Fig. 1–2 with HYSPLIT model three-day backward three-dimensional Lagrangian air parcel trajectories started from 50 m a.g.l. at geographical locations along the railway with 1 h time intervals covering the total time duration of the plume crossing events. The trajectories are colour coded in gray scale according to approximate time of air transport from areas of possible emission sources to the points

of observation, with the time stamps along the trajectories shown with black circles at 12 h intervals. In Fig. 3a and Fig. 4a we show geographical longitudes of the train in the tick labels for the upper X axis to relate location of MODIS active fires in Fig. 1–2 to the measured concentrations.

The analysis of Fig. 1–2 and Fig. 3–4 supplemented by examination of dairy records allowed us to conclude that the smoke measured within F1 and F2 plumes has originated from multiple small active fires that burned directly near the railway. Thus, we may confidently assume that the measured smoke characterizes the original emissions with negligible transformations of the constituents, and the measurements can be used to derive emission ratios. For details, please refer to the revised manuscript.

**Changes.** Fig. 1–2 are substantially edited. Discussion of Fig. 1–2 is added to the "Plume crossing episodes" section which therefore was substantially extended.

In the section "Methods of data analysis", definitions of normalized excess mixing ratios and emission ratios are added to the description of equation (1).

In Fig. 3a and Fig. 4a we show geographical longitudes of the train in the tick labels for the upper X axis to relate location of MODIS active fires in Fig. 1–2 to the measured concentrations as discussed in "Results and discussion" section in the following paragraphs:

"Beyond the data segments corresponding to the train stops..."

"In the remaining parts of the plumes..."

Further, in the discussion of $ER_{NOx/CO}$ variations: "Nevertheless, from Table 5 we see that the estimated average $ER_{NOx/CO}$ are very stable within each plume, thus indicating a similar photochemical "age" of the two plume segments in each plume. The analysis of Fig. 1–2 and Fig. 3–4 above showed that the peak excess levels of the biomass burning products measured in the F1 and F2 events have originated most probably from fires located in the vicinity of the measurement route. Therefore, we

can safely assume that all (not the peaks only) the measurements used to derive ERs in our study are heavily dominated by smoke from fresh fire plumes with a negligible average effect of chemical transformations."

In the "Conclusions" section:

"The analysis of MODIS active fire detections and HYSPLIT backward trajectories, accompanied by visual observations of many smoldering fires near the train route, shows that the excess levels of the biomass burning products measured within the plumes in the present study refer to a fresh fire smoke with negligible average effect of chemical transformations. Consequently, the estimated ERs can be safely assumed to characterize the initial chemical composition of wildfire emissions."

"The derived gas ERs are generally stable within the plumes, with the differences between the ERs estimated for different plume segments being statistically insignificant, which supports the general idea of a common fire smoke age throughout each plume, as well as a negligible effect of the changing environment on the measurements."

"The uncertainties in the ER estimates are associated mainly with variability of wildfire emissions (combustion phase, nitrogen content in the fuel) as well as with the choice of the regression approach as different assumptions on independent variables inevitably affect the final statistical inference. Chemical transformations (photochemical loss of $NO_X$ and oxidation of NMHC) of the initial wildfire emissions during their transport to the measurement route seem to have no effect on the reported average ERs and their uncertainties because of the proximity of fire emission sources to the TROICA route."

**Comment: Treatment of observations.** P10, L3-7: "The observed strong scattering of some data subsets is clearly attributable to highly complex measurement environment and the supposed strong spatial heterogeneity of the emission sources contributed to the smoke plumes. Consequently, we exclude from the analyzes the measurements producing extremely high or low dY/dX values to make our final estimates more robust with respect to various disturbing factors."

This is not an appropriate manner to handle the data. One cannot simply toss data points because they introduce scatter and reduce the correlation coefficient and increase the uncertainty of the slope in the assumed relationship. The authors should have an objective criteria for identifying data segments that are treated as the biomass smoke plume. Rejection of observations taken within the biomass plume should only be rejected using a clear, objective criteria that is based on sound reasoning – e.g. a significant influence of a local anthropogenic, instrument malfunction, or failed calibration.

**Response.** You are certainly right. To our experience, a significant influence of local (anthropogenic or biogenic) emissions is the main cause of short-term (several minutes long) fluctuations in the analyzed data sets because the events related to an instrument failure or calibration are recorded in the dairy and filtered out first at a data quality control stage. Thus we had to explain the outliers this way. The dY/dX criteria is just a technical approach to filter out fluctuations caused by non-fire sources using programmable scripts. Thus "various disturbing factors" means various local non-fire sources. Thus, we remove the lines cited above and add discussion of possible non-fire contamination.

**Changes.** In the "Results and discussion" section, we add the following paragraphs:

"Before the top of the ridge (02:50–04:00 UTC)..." and till the end of the paragraph.

"In Fig. 4a two broad $CO_2$ peaks in the western F2 plume part are observed..."

**2  Specific Comments**

**Comment.** P3, L23-24: "Both the plumes were observed in Transbaikalia – a mountainous area in the south Siberia east to the Lake Baikal known for its severe wildfire activity during warm seasons which start early in spring due to exceptionally dry

weather conditions" This sentence is awkward and I do not understand the last portion.

**Response and Changes.** We have divided this sentence into two: "Both the plumes were observed in Transbaikalia – a mountainous area in the south Siberia east to the Lake Baikal known for its severe wildfire activity. Due to dry weather conditions during winter, fire season in the region usually starts early in spring and can last from April to October."

**Comment: Measurements and instrumentation.** Grimm calibration PM3 was measured by light scattering which depends in part on the particle size distribution, chemical composition, and morphology. Please clarify if the PM3 mass density reported is based on the instrument's factory calibration or if it was calibrated for biomass burning aerosols (Aurell and Gullett, 2013; Yokelson et al., 2007) and (Nance et al., 1993). If the instrument's factory calibration was used do you anticipate any systemic bias for biomass smoke aerosols?

**Response.** We kindly thank Referee 2 for this notice. We then admit some bias due to the lack of calibration. We've added information about the $PM_3$ instrument calibration in to the "Measurements and instrumentation" section and revised interpretation of the observed $PM_3$ variability in the "Results and discussion" section. For details, please refer to the revised manuscript.

**Changes.** In the "Measurements and instrumentation" section:

"To measure $PM_3$, the Dust Indicator and Tunnel System (model 1.411), designed by GRIMM Corporation (Germany), was used. This instrument was calibrated by nephelometer PHAN-A (photoelectric photometer for aerosols) produced in Russia and calibrated by the manufacturer using the methods which are state-approved in Russia (Kopeikin et al., 2008). Calibrations were performed immediately before and after each train route. To perform the calibration, synchronous measurements by both the instruments were made during approximately 1 month both in urban and rural regions. The proper zero and span coefficients were obtained and then applied to recalculate the

**[ACPD](ACPD)**
measurements made along the train route. Such the calibration include a wide range of aerosol types from various sources which might partly compensate a possible systematic bias in the measurements of biomass smoke aerosol due to specific particle size distribution, chemical composition, and morphology which may influence the $PM_3$ mass density measured by light scattering (Aurell et al., 2013; Yokelson et al., 2007; Nance et al., 1993)."

In the "Results and discussion" section:

"The estimated $ER_{PM3/CO}$ varies within 320–385 $\frac{\text{ng m}^{-3}}{\text{µg m}^{-3}}$ with the relative uncertainties of 4–8% caused mainly by variability in the measured concentrations which, in turn, may come either from natural variability of fire emissions or from aerosol specific measurement errors. The latter are most probably related to the specific features of biomass smoke aerosol which were not completely accounted for during the instrument calibration as pointed above."

"...the $ER_{PM3/CO}$ which varies by 50–55 $\frac{\text{ng m}^{-3}}{\text{µg m}^{-3}}$ within each plume. The latter may be due to the incomplete calibration of the $PM_3$ measurement instrument for biomass smoke aerosol as pointed above, therefore we may suggest to use the average $ER_{PM3/CO}$ for each plume (which is about $360 \pm 30$ $\frac{\text{ng m}^{-3}}{\text{µg m}^{-3}}$ for F1 and $350 \pm 32$ $\frac{\text{ng m}^{-3}}{\text{µg m}^{-3}}$ for F2) to address this issue."

**Comment: NMHC detection of OVOC.** In biomass smoke a significant fraction of VOC are oxygenated-VOC (OVOC) (Akagi et al., 2011; Gilman et al., 2015). Please comment on the sensitivity of the study's NMHC detection method to OVOC, in particular the possible under-sampling of these compounds, e.g. Trabue et al., 2013.

**Response.** Thank you very much as you drew our attention for an interesting phenomenon. Indeed, burning of different substances (like OVOCs) in FIDs of different constructions used for NMHC detection may differ somewhat. So, the measured NMHC concentration may depend on VOC composition (and in particular on OVOC fraction)

[Figure]

as it was pointed out by Trabue et al. Although, our measurements of some oxygenated VOCs (acetic acid, acetone, ethanol, methanol, methacrolein, methyl-vinyl-ketone) during TROICA campaigns performed with PTR-MS (see, for example, Timkovsky et al. (2010)) showed that concentration of all these compounds are within few ppb. Concentrations of NMHC generally reach hundreds of ppb (see Table 3 and Fig. 3–4 in the manuscript) that is two orders more. The accuracy of NMHC analyzer Horiba APHA-360 is 2%, while total variability of the measurements in the analyzed fire plumes is higher. So, to our opinion, the influence of oxygenated VOCs on the NMHC analyzer readings in our case is not significant.

**Changes.** In the "Measurements and instrumentation" section, see the following paragraph:

"Previous studies show that a significant fraction of volatile organics in a biomass smoke are oxygenated compounds (OVOC)..."

**3   Results and discussion**

**Comment.** Please describe how the smoke plume boundaries were identified /selected. Were they selected based on $PM_3$ level, coincident increases in $PM_3$ and $CO$, or some other criteria?

**Response.** The smoke plume boundaries were selected as the segment with coincident and pronounced increases in ALL the measured compounds well correlated with each other (see P9,L4-8 in the discussion paper). Hence, high correlation of a measured specie with $CO$ generally means high correlation with each other measured specie within the plume. Low background concentrations of the compounds before and after the plume (see Table 3 and Fig. 3–4) suggest the absence of large sources interfering with biomass burning (the exceptions are now described in the text, see response to the next comment).

**Comment.** How did the authors assign observations to the different plume segments? Do the plume segments, e.g. F1-1 and F1-2, correspond to different stretches of the sample path? Please clarify.

**Response.** Yes, they do. The different plume segments were initially selected on the basis of varying correlations between excess levels of the major biomass burning products, $CO$ and $CO_2$ (see Table 6 and P9,L24-26 in the discussion paper). Please refer to the revised text.

**Changes.** In the "Results and discussion" section:

"Variations of the excess levels of all the measured gases and particulate matter are generally well correlated with each other within the plumes, thus supporting the notion on their common emission source. The few exceptions are discussed further."

Then, see the following paragraphs:

"Thus, Fig. 3 shows a distinct decrease in all excess mixing ratios..."

"In Fig. 4a two broad $CO_2$ peaks in the western F2 plume part are observed..."

"Given all of the above, one can see that the continuously changing environment of the measurements from the moving platform results in appreciable variations in excess mixing ratios and correlation between the major biomass burning products, $CO$ and $CO_2$ (as well as between $CO_2$ and other measured compounds that are correlated with $CO$ in this study). These variations, associated with changing surface heights in a mountainous region, as well as with non-fire emission sources as shown above, interfere with the fluctuations in the measured concentrations attributed to local forest fire emissions. To deal with the heterogeneity in the measurements conditions, we split each of the F1 and F2 plume crossing episodes into two consequent time intervals (parts, or segments, see Table 4) for further analysis according to the observed differences in excess mixing ratios and the rate of correlation between $CO$ and $CO_2$."

**Comment.** The different plume segments need to be identified on Fig. 3–4.

**Response.** The suggestion is reasonable but Fig. 3–4 do already contain much information, therefore we do not like to overload them. Thus the reader may identify the different plume segments with the UTC time stamps listed in Table 4 using the X axis in Fig. 3 and 4.

**Comment.** P9,L9-11: Simpson et al. (2011) data show dNO2/dNOx of about 70%.

**Response.** Sorry, but we can not find such a value in the cited publication. Their Table 1 suggests a "plume average" of about 88% which is 100*(1228-173)/(182-40 + 1228-173) if appropriate.

**Comment.** P9,L13-17 and Fig. 3–4 Do the "train stops" regions highlighted at the top of the plots correspond to regions excluded from the analysis?

**Response.** Thanks for your attention. Please see P5, L28-29 in the discussion paper: "The measurements during extra events (oncoming trains, tunnels, populated areas along the road) according to the records in the dairy were not used in the analysis." The train stops generally occur within the populated areas therefore the corresponding data segments were excluded as suspected for anthropogenic contamination. We've added train stops into the list of excluded events in the revised text.

**Changes.** In the "Measurements and instrumentation" section:

"Thus, the measurements during extra events (oncoming trains, tunnels, populated areas along the road, train stops) according to the records in the diary are not used in the analysis."

**Comment.** P9, L15-17: Please explain how/why these criteria for identifying anthropogenic contamination were selected.

**Response.** The $NO_X$ thresholds were selected to filter out short-term (several minutes long) peaks in $NO_X$ measurements, which are most likely associated with local anthropogenic emissions, according to our experience of the analysis of TROICA measurements. The CO threshold was selected to filter out the measurements made during

the train stop at the railway station within the rural settlement (according to the records in the dairy). There is only one episode in each plume which satisfy all these criteria (about 05:30 UTC in F1 and about 02:30 UTC in F2) and we have rejected them as suspicious.

**Comment.** Fig. 3–4. Do the dashed background lines correspond to the plume sample period? Please clarify. Fig. 3–4 should be plotted with local time or note the offset in the caption.

**Response.** Yes, they do. The offset of 8 h (see Table 1) is now in the caption.

**Changes.** In the "Results and discussion" section:

"Time series of gas mixing ratios and particle mass concentrations measured within F1 and F2 forest fire plumes are shown in Fig. 3–4 along with the estimated background levels of the measured species plotted for the period of plume crossing."

Fig. 3–4 captions are changed.

**Comment.** P9, L19: I assume "500 to 800 m a.g.l." should be "500 to 800 m a.s.l." i.e. meters above sea-level. Tables 5–6 should be merged.

**Response.** You are certainly right. Thank you.

**Comment.** P10, L1-3: $NO_X$ and BC are associated with flaming combustion and may correlate better with $CO_2$. Did the authors check for correlation vs. $CO_2$ and if so how does it compare with that vs. CO?

**Response.** It is reasonably to expect that. However, correlation with $CO_2$ is lower than with CO for ALL the measured species. That is why we choose CO as a reference specie (P6, L13-16 in the discussion paper). Thus, outliers are really outliers. The high $NO_X$ are most likely associated with local anthropogenic emissions and therefore are excluded. Other outliers that were excluded from the analysis are also suspected for contamination by non-fire sources. We've added description of the episodes of possible

contamination by non-fire emissions within the F1 and F2 fire plumes as follows.

**Changes.** Thus, in the "Results and discussion" section, we remove the following lines:

"... although a limited number of outliers persist for each particular data group with the largest scattering observed for..." and till the end of paragraph.

Instead, please refer to:

"The highest concentrations in F1 and F2 events were measured during the train stops at railway stations..." and till the end of the paragraph.

"Thus, Fig. 3 shows a distinct decrease in all excess mixing ratios..." and till the end of the paragraph.

"In Fig. 4a two broad $CO_2$ peaks in the western F2 plume part are observed..." and till the end of the paragraph.

**Comment.** P10, L3-7: "The observed strong scattering of some data subsets is clearly attributable to highly complex measurement environment and the supposed strong spatial heterogeneity of the emission sources contributed to the smoke plumes. Consequently, we exclude from the analyzes the measurements producing extremely high or low dY/dX values to make our final estimates more robust with respect to various disturbing factors."

This is not an appropriate manner to handle the data. One cannot simply toss data points because they introduce scatter and reduce the correlation coefficient and increase the uncertainty of the slope in the assumed relationship. The authors should have an objective criteria for identifying data segments that are treated as the biomass smoke plume. Rejection of observations taken within the biomass plume should only be rejected using a clear, objective criteria that is based on sound reasoning – e.g. a significant influence of a local anthropogenic, instrument malfunction, or failed calibration.

[Figure]

Also, it is unclear what is meant by: ". . .more robust with respect to various disturbing factors"

**Response.** You are certainly right. To our experience, a significant influence of local (anthropogenic or biogenic) emissions is the main cause of short-term (up to several minutes long) fluctuations in the analyzed data sets because the events related to an instrument failure or calibration are recorded in the dairy and filtered out first at a data quality control stage. Thus the dY/dX criteria is just a technical approach to filter out fluctuations caused by non-fire sources using programmable scripts. Thus "various disturbing factors" means various local non-fire sources.

**Changes.** See the changes related to the previous comment.

**Comment.** P10, L8-16: I suspect a portion of the plume F2-2 was influenced by a biogenic $CO_2$ source. Examination of Fig 6d and Fig 4a leads me to believe that F2-2 corresponds to the second portion of the plume around 3:30 to 5:30 UTC, which exhibits to broad peaks in $CO_2$ between 4:00 and 5:30 UTC for which there is not coinciding response in the CO. Additionally, the $NO_X$ does not show not increase during these broad $CO_2$ peaks (Figure 4c). Since $NO_X$ is associated with flaming combustion one would expect it to correlate with $CO_2$. Since it does not, this is further evidence that the $CO_2$ mixing ratio sampled during this plume stretch is noticeably influenced by a non-fire source. Also, the $\Delta NO_X/\Delta CO$ ratio for F2-1 and F2-2 are the same within uncertainties ($2.8 \pm 0.2$ versus $3.1 \pm 0.4$). If the source of plume segments F2-1 and F2-2 was really a fires with MCE of 0.91 and 0.97, respectively, one would expect a difference in $\Delta NO_X/\Delta CO$. I strongly disagree with the authors' interpretation of Figure 7b. It appears that dBC/dPM3 are very similar for F2-1 and F2-2. What are the plume segment average values for these ratios? I find it difficult to believe they are significantly different. In fact, I interpret Fig 7b as evidence that segments F2-1 and F2-2 originated from fires with very similar MCE. The authors should consider the $CO_2$ during this stretch to be highly suspect and not report $\Delta CO/\Delta CO_2$ or MCE for this segment.

[Figure]

**Response.** Thank you for this suggestion. The F2-2 indeed corresponds to the second portion of the F2 plume (F2-2, 03:40–5:20 UTC, see Table 4). The plume segment MEDIAN values (based on 5 min concentrations) for dBC/dPM are about 2.6% for both F2-1 and F2-2 but the scattering is strong. Thus we remove Fig. 7 and its discussion from the manuscript. We also do not report dCO/dCO2 and MCE for F2-2 following your recommendation.

We've also reconsidered the whole F2-2 part of the data and clarified the discussion of the variability in the measurements. According to the dairy records, during 04:00–04:20 UTC and 05:00-05:10 UTC, when the $CO_2$ peaks were observed and not correlated with CO, the train passed through a town and a rural settlement, respectively. Such the passage in not always associated with elevated measured concentrations, but it seems to be the case for the considered event. Thus we suggest a contribution from anthropogenic emissions into the measurements of $CO_2$, $CH_4$, NMHC, and $NO_X$ within the F2-2 plume part. For details, please refer to the paragraph in the revised manuscript cited below.

**Changes.** In the "Resulst and discussion" section:

Fig. 7 with scatter plots of BC vs. NOx and MCE vs. dBC/dPM3 and its discussion are deleted. Instead, a paragraph is added:

"In Fig. 4a two broad $CO_2$ peaks in the western F2 plume part are observed..." and till the end of the paragraph.

**Comment.** P10, L32–P11, L2: Based on my comments, I do not believe F2-2 should be considered flaming. I would limit comparison to F1-2 and F2-1, since these have valid MCE.

**Response.** We agree and revised discussion following your recommendations (see the response to the previous comment).

**Comment.** P11, L14-16: The authors have not demonstrated the sampled plumes are

likely less than 1 day old (see general comments).

**Response.** Now refer to the revised "Resulsts and discussion" section:

"Atmospheric $NO_X$ is also prone to higher variability compared to..." and till the end of paragraph. Please also see the response to the general comment related to plume age.

**Comment.** P13, Ln 27-29: I believe these were not included in Akagi et al. (2011) as they did not measure "fresh smoke" samples or the smoke age was uncertain.

Yes, it is true for the studies of Kondo et al. (2011) and Warneke et al. (2009). Meanwhile, Paris et al. (2009) reports the measurements in two Siberian plumes of 1 day old (on the basis of FLEXPART model estimates). Pirjola et al. (2015) reports the measurements of fresh emissions from an experimental fire but this study certainly could not be considered by Akagi et al. (2015) just because of the date of publication. Anyway, as far as we removed the discussion of most of the estimates from the lower left part of the plot in Fig. 7d (of the revised manuscript) following your recommendations, the following lines are abundant and we remove them as well: "The "flaming" $ER_{CO/CO2} = 2.8 \pm 0.6\%$ from the present study falls within the range of low values $ER_{CO/CO2} = 1.5 - 6\%$ in Fig. 7d. The latter corresponds to the most recent works (Paris et al, 2009; Warneke et al., 2009; Kondo et al., 2011; Pirjola et al., 2015) that apparently were not included in the compilation of Akagi et al. (2011)."

**Comment.** P16, Ln32-33 While the authors report the train operator observed some fire activity, they are clear in stating that the plumes sampled likely resulted from multiple fires, all of which were not observed. Therefore, the authors cannot relate their measured MCE to any specific observed combustion type. I agree that visual observations of fire behaviour tend to be a poor metric for classifying combustion type and MCE, especially since both flaming and smoldering typically occur simultaneously for naturally burning forest fires. However, given that EF for many species are correlated with MCE, it does have utility for extrapolating measured EF to other fire types with

different MCE regimes.

**Response.** As we understand, you reason that if visual observations of fire state do not agree with MCE – the problem is with visual observations but not with MCE. We agree. Although, visual classification of combustion regimes seems to be usual in studies that use experimental fires. We'd like to point that such the classification may give confusing results (for example, see P12, L1-4 and P12, L7-11 in the discussion paper). Therefore we have revised this part of the "Conclusions" section as follows.

**Changes.** In the "Conclusions" section:

"The authors did not find any definite relation between the visually observed combustion type (smoldering or flaming) and MCE values neither in this study, mainly because of the lack of detailed information on fire state, nor in the previous studies where emissions from experimental fires were attributed to flaming or smoldering combustion on the basis of visual inspections (Cofer et al., 1998; Pirjola et al., 2015). Thus, we are cautious in using visual observations to attribute fire emissions to a specific combustion type since both flaming and smoldering typically occur simultaneously for naturally burning forest fires."

**4 Comparison with other published results**

**Comment.** The discussion and figures are a bit confusing. The authors seem to include studies where the plumes sampled were older than 1 day and therefore are not emission ratios and not appropriate for comparison with the current work. I strongly recommend the authors limit the comparison to studies where the plume samples were <= 1 day old and result from boreal fires.

**Response and Changes.** This also seems reasonably. Although, there are only three publications (known to the authors of the present study) that report both BC and CO

emissions for boreal fires. Therefore, we cite them all, with a warning about differences in plume age. And in Fig. 9b in the discussion paper, there is no clear relation between the plume age and $ER_{BC/CO}$, with ER from Paris et al. (2009) for plume of 1 day old being close ERs from Kondo et al. (2011) and Warneke et al. (2009) for boreal fire plumes of several days old. Following your recommendation, we remove discussion of $ER_{CO/CO2}$ from Kondo et al. (2011) and Warneke et al. (2009) but we'd like to keep the $ER_{BC/CO}$ from these publications.

**Comment.** NMHC comparison and Figure 8b: The NMHC EF based on Laursen et al. (1992) and Urbanski et al. (2009) are the sum of only a handful of compounds and not comprehensive VOC measurement like that constructed in the current study. This should be clarified in the text.

**Response and Changes.** We agree. Although it does not decrease the value of the data for the comparison. See in the revised text: "Thus, Laursen et al. (1992) and Urbanski et al. (2009) report the measurements of a very limited number of individual NMHC compounds which can not be directly compared to the comprehensive NMHC measurements employed in the present study but are shown in Fig. 7b because of the deficit of NMHC observations in boreal forest fire plumes."

**5  Technical Corrections**

**Comments.** The authors should define chemical formulas when first introduced. P1, L16: Insert "the" between (btw) "and" and "boreal"

P1, L16: change "became" to "become"

P1, L18: insert "the" btw "including" and "global"

P2, L3: Insert "the" btw "In" and "future"

P2, L14: Change "OH-" to "OH" it's a radical not an ion. No charge.

P2, L17: change "is" to "are"

P2, L19: change "on the basis" to "by"

P2, L21: delete ",Canada, and Alaska as a ...."

P3, L1: insert "of' btw "all" and "these"

P3, L17: change "substantia amount" to "many"

P3, L32: insert "that" before "originated".

There are many similar errors in English usage throughout the remainder of the manuscript that need correction.

**Response.** Done, thanks.

**Comment.** Figure 3a – The CO and CO2 background lines have wrong colors

**Response.** Thanks for your attention. It should be ok now.

**Comment.** Table 6 change "PM1" to "PM3"

**Response.** Table 6 is merged with Table 5 according to your recommendation.

**Comment.** Fig. 8–9: The plotted symbols do not all match the legend, Vasileva et al., 2017 and Pirjola et al., 2015 are different.

**Response and Changes.** We removed the results of Kondo et al. (2011) and Warneke et al. (2009) from Fig. 7 of the revised manuscript following you suggestion. The legend and symbols in Fig. 7 and Fig. 8 are rearranged. Although, note that ig. 7 and Fig. 8 have independent legends, and the symbols are used independently.

**References**

[revised manuscript text omitted]